# OTUD6 deubiquitination of RPS7/eS7 on the free 40 S ribosome regulates global protein translation and stress

Sammy Villa [1,9], Pankaj Dwivedi [2,10], Aaron Stahl[3,4], Trent Hinkle [2], Christopher M. Rose [2], Donald S. Kirkpatrick[2,11], Seth M. Tomchik [3,4,5,6], Vishva M. Dixit [7] & Fred W. Wolf [1,8] ✉

Ribosomes are regulated by evolutionarily conserved ubiquitination/deubiquitination events. We uncover the role of the deubiquitinase OTUD6 in regulating global protein translation through deubiquitination of the RPS7/eS7 subunit on the free 40 S ribosome in vivo in *Drosophila*. Coimmunoprecipitation and enrichment of monoubiquitinated proteins from catalytically inactive OTUD6 flies reveal RPS7 as the ribosomal substrate. The 40 S protein RACK1 and E3 ligases CNOT4 and RNF10 function upstream of OTUD6 to regulate alkylation stress. OTUD6 interacts with RPS7 specifically on the free 40 S, and not on 43 S/48 S initiation complexes or the translating ribosome. Global protein translation levels are bidirectionally regulated by OTUD6 protein abundance. OTUD6 protein abundance is physiologically regulated in aging and in response to translational and alkylation stress. Thus, OTUD6 may promote translation initiation, the rate limiting step in protein translation, by titering the amount of 40 S ribosome that recycles.

Regulation of protein translation ensures that cells and organisms adapt to changing conditions, including nutrient availability, chemical and environmental stressors, cell proliferation or quiescence, and aging, as well as providing mechanisms to adapt to gene-level differences in translation conditions[1]. Upregulated global protein translation is also a hallmark of many cancers[2]. The basic machinery of protein translation and the amino acid sequence of core ribosomal proteins are profoundly conserved evolutionarily; many regulatory mechanisms are also conserved. Protein translation regulation can occur globally and at the level of individual RNAs. Global regulation of translation is achieved by two well-characterized regulatory steps on specific effectors of translation initiation[3]. First, phosphorylation of the translation initiation factor eIF2α in response to cellular stressors

blocks eIF2 association with initiator methionyl-tRNA to hinder formation of the 43 S translation preinitiation complex and 5'-untranslated region scanning, reducing protein translation. Second, 4E-BP factor phosphorylation releases eIF4E to promote cap-dependent mRNA translation under conditions of nutrient availability.

Ubiquitination is a mechanism for the regulation of protein translation. The small protein ubiquitin is added to lysine side chains of target proteins through a three step process that is commonly completed by one of hundreds of E3 ubiquitin ligases that determine the substrate for ubiquitination[4]. A smaller but substantial number of deubiquitinases can reverse or edit protein ubiquitination[5]. Ubiquitination, either as monoubiquitination or in forming polyubiquitin chains, can target substrate proteins for degradation or alter their

[1]Quantitative and Systems Biology, University of California, Merced, CA 95343, USA. [2]Department of Microchemistry, Proteomics, and Lipidomics, Genentech Inc., 1 DNA Way, South San Francisco, CA 94080, USA. [3]Neuroscience and Pharmacology, University of Iowa, Iowa City, IA 52242, USA. [4]Department of Neuroscience, Scripps Research, Jupiter, FL 33458, USA. [5]Iowa Neuroscience Institute, University of Iowa, Iowa City, IA 52242, USA. [6]Stead Family Department of Pediatrics, University of Iowa, Iowa City, IA 52242, USA. [7]Department of Physiological Chemistry, Genentech, 1 DNA Way, South San Francisco, CA 94080, USA. [8]Department of Molecular and Cell Biology, University of California, Merced, CA 95343, USA. [9]Present address: Calico Life Sciences, 1170 Veterans Boulevard, South San Francisco, CA 94080, USA. [10]Present address: Merck, West Point, PA 19486, USA. [11]Present address: Xaira Therapeutics, Brisbane, CA 94005, USA. ✉e-mail: fwolf@ucmerced.edu

physical interactions with other molecules[6]. Cellular stressors and complications in protein translation trigger ubiquitination events on ribosomes, setting in motion molecular pathways to halt the production of aberrant proteins. Examples include the extensive use of ubiquitin modifications on ribosome subunits in ribosome quality control pathways and in the handling of codon optimality during translation elongation[7–9]. Aspects of normal translation initiation and elongation are also regulated by highly transient ubiquitination events. Both ubiquitination and deubiquitination events are subjects of regulation, providing layers of control over protein stability and function.

Here, we characterize the function of the OTU-class deubiquitinase OTUD6 in *Drosophila*. We discovered that OTUD6 protein levels bidirectionally regulate global protein translation levels by deubiquitinating monoubiquitinated RPS7 on the free 40 S ribosomal subunit. This pathway also promotes resistance to alkylation stress. OTUD6 is orthologous to OTUD6A and OTUD6B in mammals and to OTU2 in yeast. Naturally occurring OTUD6B mutations in humans are

associated with intellectual disability and dysmorphic craniofacial features[10]. OTUD6B is upregulated in many cancers, and it can promote cell proliferation[11–13]. Yeast OTU2 was recently discovered to deubiquitinate the RPS7 ortholog eS7A[14]. We propose that regulation of OTUD6 protein abundance is a mechanism for setting the rate of global protein translation by metering the availability of free 40 S ribosomal subunits for the formation of translation initiation complexes.

## Results

### OTUD6 catalytic activity is essential for resistance to alkylation and oxidation stress

To study OTUD6 in *Drosophila*, we created at the endogenous locus two catalytically inactive mutants, a null mutant, and wild-type and catalytically inactive FLAG.HA-tagged OTUD6 (Fig. 1a). Both catalytically inactive mutants, $OTUD6^{C183A}$ and $OTUD6^{C183R}$ where the catalytic cysteine was replaced with alanine or arginine, were homozygous

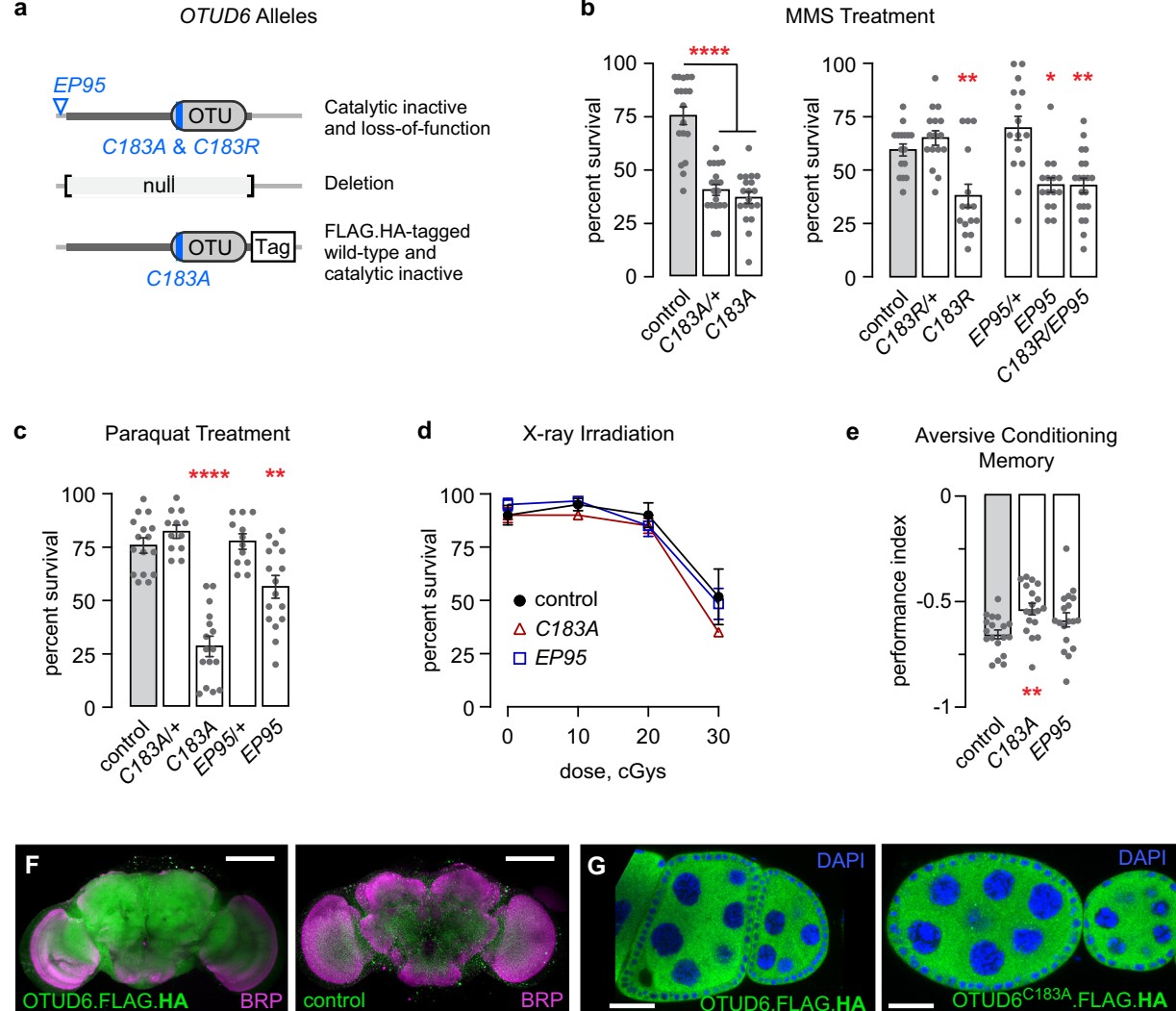

**Fig. 1 | OTUD6 promotes resistance to alkylation and oxidation stress.**
**a** *Drosophila OTUD6* mutations and tagged endogenous forms. **b** OTUD6 catalytically inactive and loss-of-function mutants are sensitive to exposure to 0.05% MMS, measured at 32 h. Each data point represents the percent survival in a vial of 15 flies. One-way ANOVA/Dunnett's, compared to control. (left: $n = 14, 15, 15$. right: $n = 16, 16, 15, 15, 16$). **c** OTUD6 catalytically inactive and loss-of-function mutants are sensitive to exposure to 10 mM paraquat, measured at 72 h. One-way ANOVA/Dunnett's, compared to control. ($n = 16, 12, 12, 12, 16$) **d**. Survival of third instar larvae exposed to X-ray irradiation. ($n = 3, 3, 3$). **e** Pavlovian short-term memory of

aversive shock - neutral odor pairing. One-way ANOVA/Dunnett's, compared to control. ($n = 18, 18, 18$). **F** OTUD6 is uniformly distributed in the *Drosophila* brain. Left: OTUD6.FLAG.HA detected with anti-HA (green). Right: untagged wild-type control. Scale bar: 50 μm. **G** Distribution of endogenously tagged OTUD6 (left) and $OTUD6^{C183A}$ (right) in ovary egg chambers. Large nuclei and surrounding cytoplasm in the center are nurse cells that are surrounded by smaller follicle cells. Scale bar: 25 μm. Data are presented as mean values +/− SEM. Dots on on bar graphs and n represent biological replicates. Source data and statistics are provided as a Source Data file.

viable with no morphological or obvious behavioral defects. *OTUD6^EP95* harbors a transposon insertion in the *OTUD6* 5′ untranslated region that resulted in near complete depletion of *OTUD6* mRNA levels in outwardly normal homozygotes (Supplementary Fig. 1A). *OTUD6^null* was lethal as a homozygote and in trans to a deficiency that uncovers *OTUD6*, but it was viable in trans to *OTUD6^CI83A* and *OTUD6^CI83R*. The *OTUD6^null* was also viable in trans to lethal mutations in each of the flanking genes. Thus, OTUD6 has an essential biological function that is independent of its catalytic activity. To identify a function for OTUD6, we subjected adult flies to a variety of chemical and environmental stressors. Both catalytically inactive and loss-of-function *OTUD6* mutants were markedly sensitive to the alkylating agent methyl methanesulfonate (MMS) (Fig. 1b). *OTUD6^CI83A*, with a predicted increased affinity for ubiquitinated targets, was also MMS sensitive as a heterozygote, while the predicted lower affinity *OTUD6^CI83R* was MMS-sensitive only as a homozygote[15]. *OTUD6^CI83R* failed to compliment *OTUD6^EP95*, indicating that protection from alkylation damage mapped to the OTUD6 locus and to its deubiquitinase function. *OTUD6* mutants were also sensitive to the oxidative stressor paraquat (Fig. 1c). Alkylative and oxidative stressors impair protein translation and damage nucleic acids[16–18]. To determine if OTUD6 regulates DNA damage repair, larvae were exposed to X-ray irradiation. There was no impact on mutant survival compared to control, suggesting that OTUD6 does not regulate DNA break repair (Fig. 1d). We also used a genetic method that reports the types of double-stranded DNA repair events[19]. With this assay we detected a small decrease in homology-directed repair in *OTUD6^CI83A* (Supplementary Fig. 1B).

To ask if *Drosophila* OTUD6 may exhibit similar functions to human OTUD6B, specifically a role in intellectual ability, we test the ability of *OTUD6* mutants to associate an odor cue with an aversive electric shock[20]. *OTUD6^CI83A* mutants showed a decrease in short term memory, whereas *OTUD6^EP95* trended towards a decrease (Fig. 1e, Supplementary Fig. 1C).

Finally, we characterized the tissue and subcellular distribution of OTUD6. High throughput -omics indicated that OTUD6 is expressed in all tissues, and that it is more highly expressed in tissues characterized by high cell proliferation and high protein translation rates, including the ovaries, testes, and imaginal discs, as well as in the central nervous system[21]. In the brain, OTUD6.FLAG.HA was uniformly distributed in all cells, and it was present in both neuronal processes and the cell bodies (Fig. 1F). OTUD6 protein was most prominent in the cytoplasm in the large nurse cells and the smaller surrounding follicle cells of the developing ovarioles, sites where protein subcellular distribution is readily assessed in *Drosophila* (Fig. 1G). OTUD6 was also present in the nucleus and the nucleolus (Supplementary Fig. 1D). *OTUD6^CI83A*.FLAG.HA had an identical subcellular distribution in the cells of the ovariole.

## OTUD6 associates with the 40 S ribosomal subunit to regulate cellular stress through deubiquitination

To identify potential OTUD6 substrates, we performed co-immunoprecipitation (co-IP) followed by mass spectrometry (MS) with OTUD6^CI83A.FLAG.HA as compared to untagged OTUD6^CI83A from *Drosophila* heads. This design allowed us to segregate OTUD6^CI83A interactors from non-specific binding to the immunoprecipitation matrix. Nearly all OTUD6^CI83A interacting proteins were components of the 40 S ribosomal subunit or of the RNA exosome (Fig. 2a). We confirmed that OTUD6^CI83A associates with a component of the 40 S ribosome, RACK1, by IP of OTUD6^CI83A.FLAG.HA followed by western detection (Fig. 2b). RACK1 is a multifunctional protein that coordinates the ubiquitination of the 40 S ribosome by specific E3 ligases, particularly in ribosome quality control (RQC) pathways[22–25]. To test if OTUD6 might regulate ribosome ubiquitination pathways, we determined if OTUD6 and RACK1 interact genetically. Two independent *Rack1* mutations had no effect on MMS sensitivity, but each mutation

suppressed *OTUD6^CI83A* MMS sensitivity (Fig. 2c). This result suggests that reduced RACK1-dependent ubiquitination of the 40 S ribosome decreases the need for OTUD6 deubiquitination activity. To identify a ubiquitination pathway or pathways for OTUD6, we tested the MMS sensitivity of *Drosophila* that were mutant for orthologs of E2 conjugating enzymes and E3 ligases that ubiquitinate the 40 S ribosome in response to oxidative stress, preinitiation complex collisions during initiation, ribosome collisions during translation elongation, and in non-functional rRNA decay (Fig. 2d)[23,26–29]. Three E3 ligases, CNOT4, RNF10, and RNF123/KPC1 suppressed *OTUD6^CI83A* sensitivity to MMS treatment (Fig. 2e-g, Supplementary Fig. 2A–G). A fourth, FBXW7/CDC4, showed weaker but significant suppression of *OTUD6^CI83A* MMS sensitivity (Supplementary Fig. 2A). All other gene mutants either did not suppress *OTUD6^CI83A* MMS sensitivity or were sensitive on their own. CNOT4 is proposed to ubiquitinate RPS7 during 80 S assembly at the start codon, and when nonoptimal codons are encountered, a process that can be coupled to mRNA stability by the CCR4-NOT complex[30,31]. RNF10 ubiquitinates RPS2/uS5 and RPS3/uS3 in RQC pathways, whereas RNF123/KPC1 is less well characterized but was shown to ubiquitinate RPS3[25,32,33]. RPS2, RPS3, and RPS7 all co-immunoprecipitated with OTUD6^CI83A (Fig. 2a). Thus, OTUD6 likely acts downstream of RNF10, RNF123, and CNOT4 to deubiquitinate one or more substrates on the 40 S ribosomal subunit.

Mutations in DIS3, the catalytic subunit of the RNA exosome and one of the proteins most significantly associated with OTUD6^CI83A, also suppressed *OTUD6^CI83A* MMS sensitivity (Supplementary Fig. 2H). We confirmed that DIS3 was associated with OTUD6^CI83A by IP-western analysis, however DIS3 showed moderate non-specific binding to the beads used for immunoprecipitation (Supplementary Fig. 2I). All ten RNA exosome subunits were enriched with OTUD6^CI83A in the IP-MS (Fig. 2a, Supplementary Data 1). The RNA exosome is a general ribonucleolytic protein complex that is highly conserved in eukaryotes[34]. The RNA exosome is recruited to the ribosome by the SKI complex[35,36]. We recovered no components of the SKI complex in our IP-MS experiment, suggesting that OTUD6-containing protein complexes may interact with the DIS3 RNA exosome in a different state.

## OTUD6 deubiquitinase activity promotes protein translation

OTUD6 association with the 40 S small ribosomal subunit prompted us to ask if protein translation was affected in OTUD6 mutants. To test this, we performed a puromycin incorporation assay. Puromycin is a tyrosyl-tRNA analog that is incorporated into elongating polypeptide chains that is easily detected on western blots. We fed puromycin to flies overnight and collected fly heads for western blotting. Both catalytically inactive OTUD6 mutants reduced puromycin incorporation by approximately 50% (Fig. 3a). Global reduction in protein synthesis generally extends lifespan[37]. We assessed lifespan in *OTUD6^CI83A* and in the loss-of-function *OTUD6^EP95*, and found that both lived dramatically longer (Fig. 3b). *Drosophila* harboring mutations that impair ribosome function or abundance typically exhibit a constellation of phenotypes classified as Minute syndrome, including reduced body size, extended development, short and thin bristles, and poor fertility[38]. There was no impact of *OTUD6* mutations on adult size, bristle morphology, or fertility. However, eclosion was delayed in *OTUD6* mutants, indicating an extended period of development (Fig. 3c). Moreover, there is no effect on developmental survival from egg to adulthood (Fig. 3d). Thus, OTUD6 deubiquitinase activity promotes protein translation, but its role is more specific than that of other protein translation factors that cause the Minute phenotype when mutant.

## OTUD6 deubiquitinates RPS7 and bidirectionally regulates the rate of protein synthesis

To determine a ribosomal substrate for OTUD6, we performed a serial polyubiquitin-monoubiquitin enrichment, followed by mass

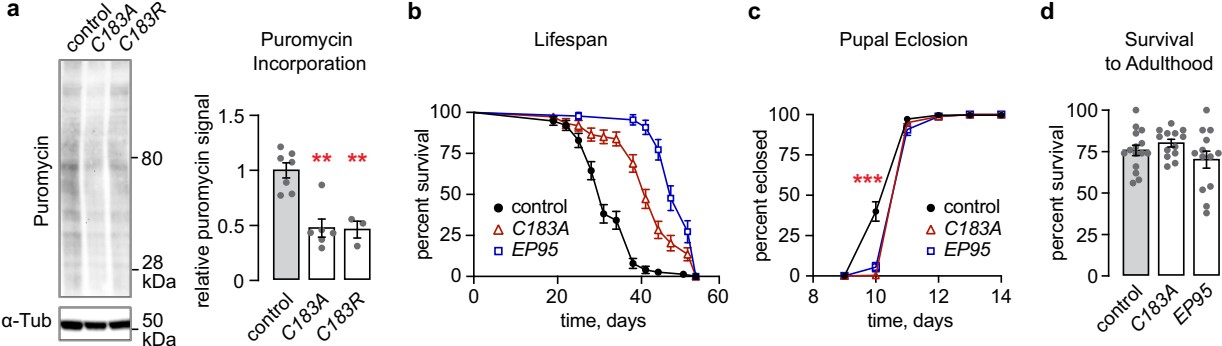

**Fig. 2 | OTUD6 physically Interacts with the 40 S ribosome and opposes 40 S ribosome ubiquitination. a** Volcano plot of OTUD6$^{C183A}$ physical interactors, $n = 3$ biological replicates per genotype. **b** FLAG co-immunoprecipitation with tagged OTUD6$^{C183A}$ followed by western analysis for RACK1 and OTUD6. **c** Genetic interaction analysis of heterozygous mutations for *OTUD6* and *Rack1*. One way ANOVA with Sidak's multiple comparisons test. ($n = 24, 24, 20, 20, 24, 24$) **d** E2 conjugating enzymes and E3 ligases tested for genetic interactions with OTUD6 for MMS sensitivity. **e–g** Suppressors of *OTUD6$^{C183A}$* MMS sensitivity. One way ANOVA with Sidak's multiple comparisons test. (**e**: $n = 15, 15, 14, 14$. **f**: $n = 20, 20, 20, 19, 13, 15$. **g**: $n = 20, 16, 17, 9$). Data are presented as mean values +/− SEM. Dots on on bar graphs represent biological replicates. Source data and statistics are provided as a Source Data file.

**Fig. 3 | OTUD6 deubiquitinase activity promotes protein translation.**
**a** Puromycin incorporation assay using *Drosophila* head lysate. Puromycin signal was normalized to tubulin. One-way ANOVA/Dunnett's, compared to control. ($n = 7, 6, 3$). **b** *OTUD6* mutants have extended lifespan. Representative graph of three independent experiments. Curves compared using log-rank (Mantel-Cox) test to control, *OTUD6$^{C183A}$* ($P < 0.0001$) and *OTUD6$^{EP95}$* ($P < 0.0001$). ($n = 75, 74, 45$). **c** Developmental time, from egg to adulthood, is extended in *OTUD6* mutants. Graph displays cumulative of eclosion to adult. One-way ANOVA/Dunnett's, compared to control at day 10. ($n = 8, 6, 6$). **d** *OTUD6* mutants have no impact on survival to adulthood. Data are presented as mean values +/− SEM. ($n = 15, 13, 13$). Dots on bar graphs represent biological replicates. Source data and statistics are provided as a Source Data file.

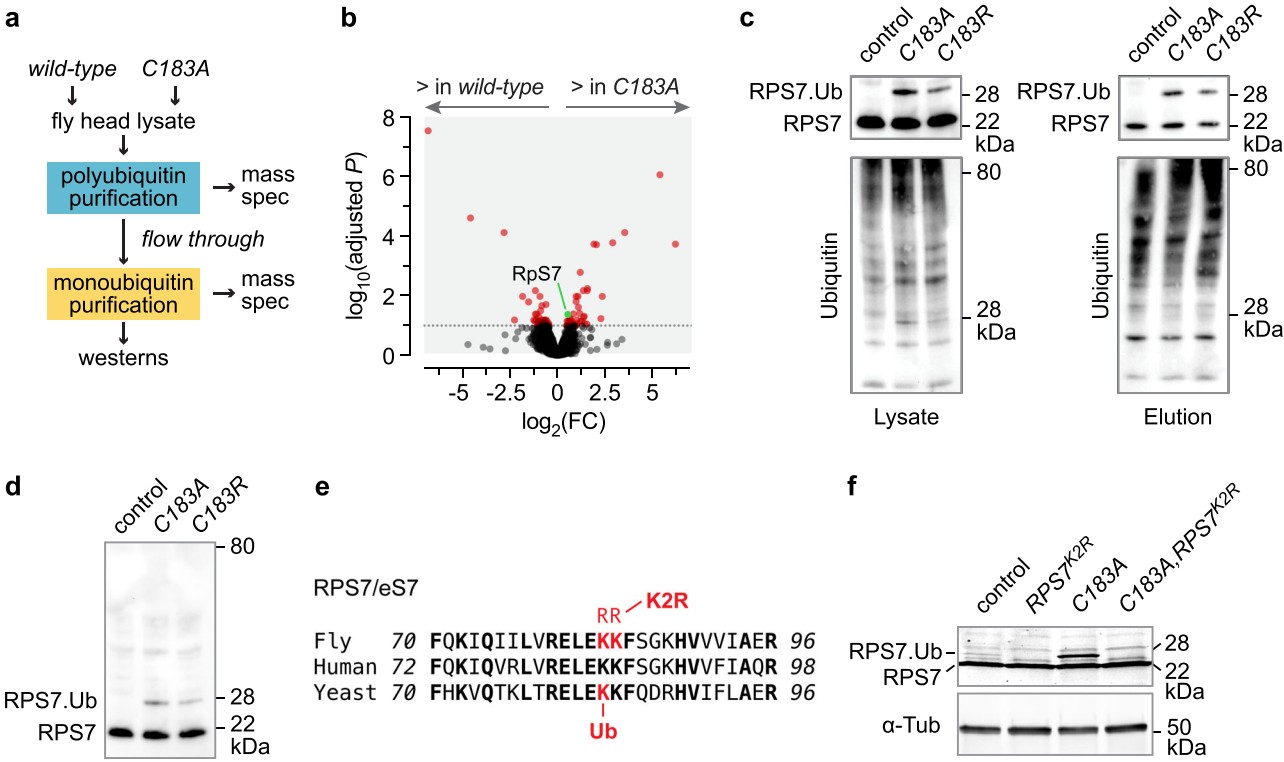

**Fig. 4 | OTUD6 deubiquitinates monoubiquitinated RPS7. a** Serial mono-ubiquitin capture scheme. **b** Mass spectrometry analysis to identify differentially enriched proteins in monoubiquitin-selective capture reagents following a poly-ubiquitin preclear, *OTUD6$^{C183A}$* compared to genetic background control. Red indicates significant hits. $n = 3$ biological replicates per genotype. See Methods for statistical analysis. **c** Catalytically inactive *OTUD6* mutants are enriched for RPS7 monoubiquitination. Serial monoubiquitin capture followed by western for RPS7 and total ubiquitin. Blots are representative of 3 biological replicates. **d** Lysate

western for RPS7 (same as **c**), indicating monoubiquitination and no poly-ubiquitination. **e** Protein sequence of RPS7 in *Drosophila melanogaster*, *Homo sapiens*, and eS7A in *Saccharomyces cerevisiae*, showing the monoubiquitination site in yeast and the two residues that were mutated to arginine, K83R and K84R, to create *Drosophila* RPS7$^{K2R}$. **f** RPS7 monoubiquitination is absent in *RPS7$^{K2R}$* and *OTUD6$^{C183A}$,RPS7$^{K2R}$* flies. Blots are representative of 3 biological replicates. Data are presented as mean values +/− SEM. Source data and statistics are provided as a Source Data file.

spectrometry identification of proteins from *Drosophila* heads of *OTUD6$^{C183A}$* compared to *wild-type* control (Fig. 4a, Supplementary Data 2 & 3). This scheme allows us to enrich for substrates that are either polyubiquitinated or monoubiquitinated in *OTUD6$^{C183A}$*. The enrichments were effective, as assayed by western analysis with FK1, a ubiquitin antibody specific for polyubiquitinated substrates, and P4D1, an antibody that recognizes total ubiquitin (Supplementary Fig. 3A, B). RPS7, a 40 S ribosomal protein, was the sole ribosomal or ribosome-associated protein to be significantly enriched in the catalytically inactive mutant (Fig. 4b, Supplementary Fig. 3C). To test if RPS7 is deubiquitinated by OTUD6, we produced a *Drosophila* RPS7 antibody, and probed *Drosophila* head lysates and serial polyubiquitin-monoubiquitin eluates. In both *OTUD6$^{C183A}$* and *OTUD6$^{C183R}$* we detected a higher molecular weight band that was consistent with mono-ubiquitinated RPS7 (RPS7.Ub) (Fig. 4c). We did not detect any additional higher molecular weight bands indicating a lack of RPS7 polyubiquitination (Fig. 4d). Treatment of head lysate with the broad spectrum deubiquitinase USP2 decreased RPS7.Ub without affecting non-ubiquitinated RPS7 levels (Supplementary Fig. 3D). We mutated two adjacent lysine residues to arginines in the endogenous *Drosophila* RPS7, to test if monoubiquitination occurred at a site that is conserved in yeast eS7A, *Drosophila* RPS7, and human RPS7 (Fig. 4e)[14]. The resulting mutant flies, *Rps7$^{K2R}$*, were homozygous viable and fertile, indicating that RPS7$^{K2R}$ protein was incorporated into the ribosome, and that ribosomal function was not adversely affected. RPS7 mono-ubiquitination was completely absent in *Rps7$^{K2R}$* single mutant and in *OTUD6$^{C183A}$,Rps7$^{K2R}$* double mutant fly lysates, and by ubiquitin immu-noprecipitation (Fig. 4f, Supplementary Fig. 3e). Thus, OTUD6

deubiquitinates monoubiquitinated RPS7 at a lysine residue that is conserved from yeast to flies.

## OTUD6 protein abundance can determine global protein translation levels and RPS7 monoubiquitination

OTUD6 deubiquitination of RPS7 may be tied to OTUD6 regulation of protein translation. To test this, we bidirectionally altered OTUD6 protein abundance, and assessed the levels of global protein transla-tion and RPS7 monoubiquitination. We decreased OTUD6 protein levels by expressing an RNAi directed against OTUD6 (*UAS-OTUD6.IR*) in all neurons (*elav-Gal4*) using the GAL4/UAS binary expression sys-tem (Supplementary Fig. 4A)[39]. Similarly, we increased OTUD6 protein levels using *elav-Gal4* to drive *UAS-OTUD6.HA (OTUD6.OE)*. Critically, OTUD6 RNAi decreased protein translation, and OTUD6 over-expression increased protein translation (Fig. 5a). Thus, OTUD6 protein levels can set the amount of protein translation. Reciprocally, OTUD6 RNAi increased RPS7.Ub, whereas OTUD6 overexpression nearly eliminated RPS7.Ub (Fig. 5b). Overall levels of RPS7 were not changed with either manipulation (Supplementary Fig. 4B). These findings strongly suggest that OTUD6 regulates the level of global protein translation through deubiquitination of RPS7: increased OTUD6 increases RPS7 deubiquitination and enhances protein translation.

## OTUD6 regulation of RPS7 monoubiquitination is critical for resilience to alkylation stress and for promoting protein translation

Three proteins that regulate ribosomal ubiquitination, when mutant, suppress *OTUD6$^{C183A}$* sensitivity to MMS, suggesting that they act

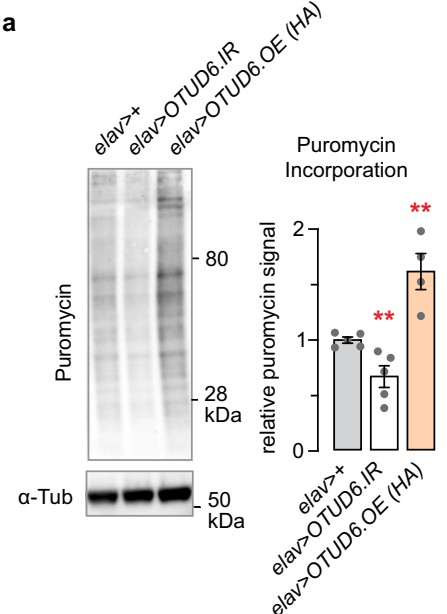

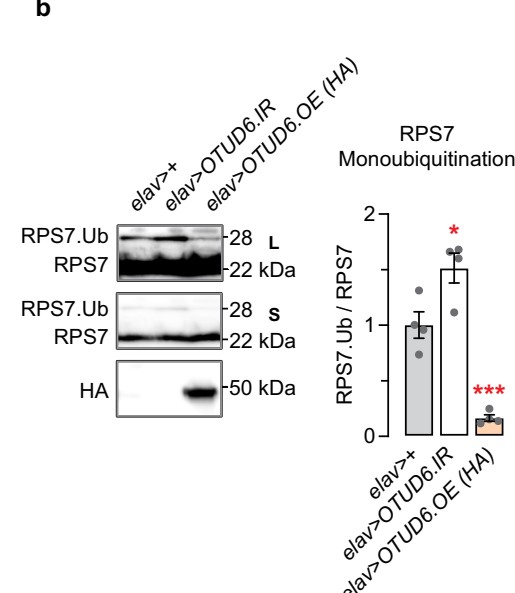

**Fig. 5 | OTUD6 bidirectionally regulates the level of protein translation and RPS7 deubiquitination. a** Pan neuronal (*elav-Gal4*) RNAi knockdown (*UAS-OTUD6.IR*) and overexpression (*UAS-OTUD6.OE*) of OTUD6 resulted in decreased and increased puromycin incorporation, respectively, in fly heads. One-way ANOVA/Dunnett's, compared to control. (*n* = 5,5,4). **b** Pan neuronal (*elav-Gal4*) RNAi knockdown (*UAS-OTUD6.IR*) and overexpression (*UAS-OTUD6.OE*) of OTUD6

resulted in increased and decreased RPS7 monoubiquitination in fly heads. Ubiquitinated RPS7 normalized to total ubiquitin. L: long exposure; S: short exposure. One-way ANOVA/Dunnett's, compared to control. (*n* = 4, 4, 4). Data are presented as mean values +/− SEM. Dots on on bar graphs represent biological replicates. Source data and statistics are provided as a Source Data file.

upstream of OTUD6 in a pathway that regulates alkylation stress (Fig. 2e–g). We tested if deubiquitination of RPS7 may be a mechanism for these genetic interactions. Flies with mutations in *Rack1*, *RNF10*, or *Cnot4* markedly reduced RPS7 monoubiquitination in *OTUD6^{C183A}* and *OTUD6^{C183A/+}* flies (Fig. 6a, Supplementary Fig. 5A–D). This ties the regulation of RPS7 monoubiquitination to alkylation stress sensitivity. It further indicates that RNF10 and RACK1 promote monoubiquitination of RPS7, directly or indirectly. The increased MMS sensitivity of flies with reduced OTUD6 function may be a consequence of increased or misregulated RPS7 monoubiquitination. As a test, we measured RPS7 ubiquitination status when flies were treated with MMS in *wild-type* control and *OTUD6^{C183A}* mutant flies. MMS treatment reduced RPS7 monoubiquitination in control and not in *OTUD6^{C183A}* flies, and monoubiquitination in MMS treated *OTUD6^{C183A}* remained markedly higher than in untreated controls (Fig. 6b, Supplementary Fig. 5E). MMS treatment did not affect overall ubiquitination in fly heads (Supplementary Fig. 5F). RPS7 is proposed to be monoubiquitined by CNOT4 in the presence of nonoptimal codons in mRNAs[7,31]. We used GFP transgenes that contain 100% common codons (GFP0D) or approximately 50% nonoptimal codons (GFP54C3') to assess the effect on RPS7 monoubiquitination[40]. We observed an increase in RPS7 monoubiquitination and no change in RPS7 or OTUD6 abundance in GFP54C3' flies (Fig. 6c). The effect was not due to changes in GFP54C3' abundance in *OTUD6^{C138A}*, suggesting that OTUD6 acts downstream of RPS7 ubiquitination. These results suggest that overexpression of a nonoptimal codon transcript in all cells engages the RPS7 monoubiquitination pathway in *Drosophila*.

We next tested the effects of complete elimination the OTUD6-RPS7 ubiquitination/deubiquitination pathway. If the effects on MMS sensitivity were due to misregulation of RPS7 monoubiquitination, then *OTUD6^{C183A}*,*Rps7^{K2R}* double mutation might restore normal sensitivity. Alternatively, a functional OTUD6-RPS7 regulatory may be needed to maintain resistance to MMS. *Rps7^{K2R}* flies were more sensitive to MMS, and the double mutants were as sensitive as the

*OTUD6^{C183A}* single mutants, suggesting the functional pathway is needed for resistance to alkylation stress (Fig. 6d). Global protein translation was reduced by about 50% in either mutant alone and in *OTUD6^{C183A}*,*Rps7^{K2R}* double mutants, suggesting that the regulation of protein translation by OTUD6 is solely through deubiquitination of RPS7 (Fig. 6e). Finally, we confirmed prior findings that protein translation is markedly reduced by MMS treatment, and we further found that there was no additive or synergistic effect on protein translation in *OTUD6^{C183A}* mutants (Supplementary Fig. 5G)[41]. MMS is reported to reduce protein translation through the integrated stress response pathway; the OTUD6-RPS7 pathway may act in parallel in a compensatory fashion[42]. Together, these results suggest that RPS7 monoubiquitination by a variety of E3 ligases, directly or indirectly, and subsequent deubiquitination by OTUD6 is critical for maintaining resilience to alkylation stress and for the promotion of protein translation. Direct ubiquitination of RPS7 by RNF10 and RACK1 is not yet demonstrated.

## OTUD6 is specifically associated with the free 40 S ribosomal subunit

To better define the steps of the ribosome cycle or ribosome biogenesis that OTUD6 is associated with, we performed co-immunoprecipitation experiments in the presence of cycloheximide, to preserve interaction of the 40 S with the 60 S in the 80 S translating ribosome while maintaining a pool of recycling ribosome subunits. First, we asked what proteins associate with wild-type and catalytically inactive OTUD6. Monoubiquitinated RPS7 copurified with tagged OTUD6^{C183A}, as expected (Fig. 7a). However, non-ubiquitinated RPS7 was undetectable, indicating that the catalytically inactive OTUD6 captures a precise step in the lifecycle of the 40 S ribosome. Similarly, we were unable to detect RPL11, suggesting that neither OTUD6 nor OTUD6^{C183A} was associated with the 80 S ribosome, where ubiquitinated RPS7 is also reported to be present[7,30]. Finally, the 43 S and 48 S preinitiation complexes also showed no association with either

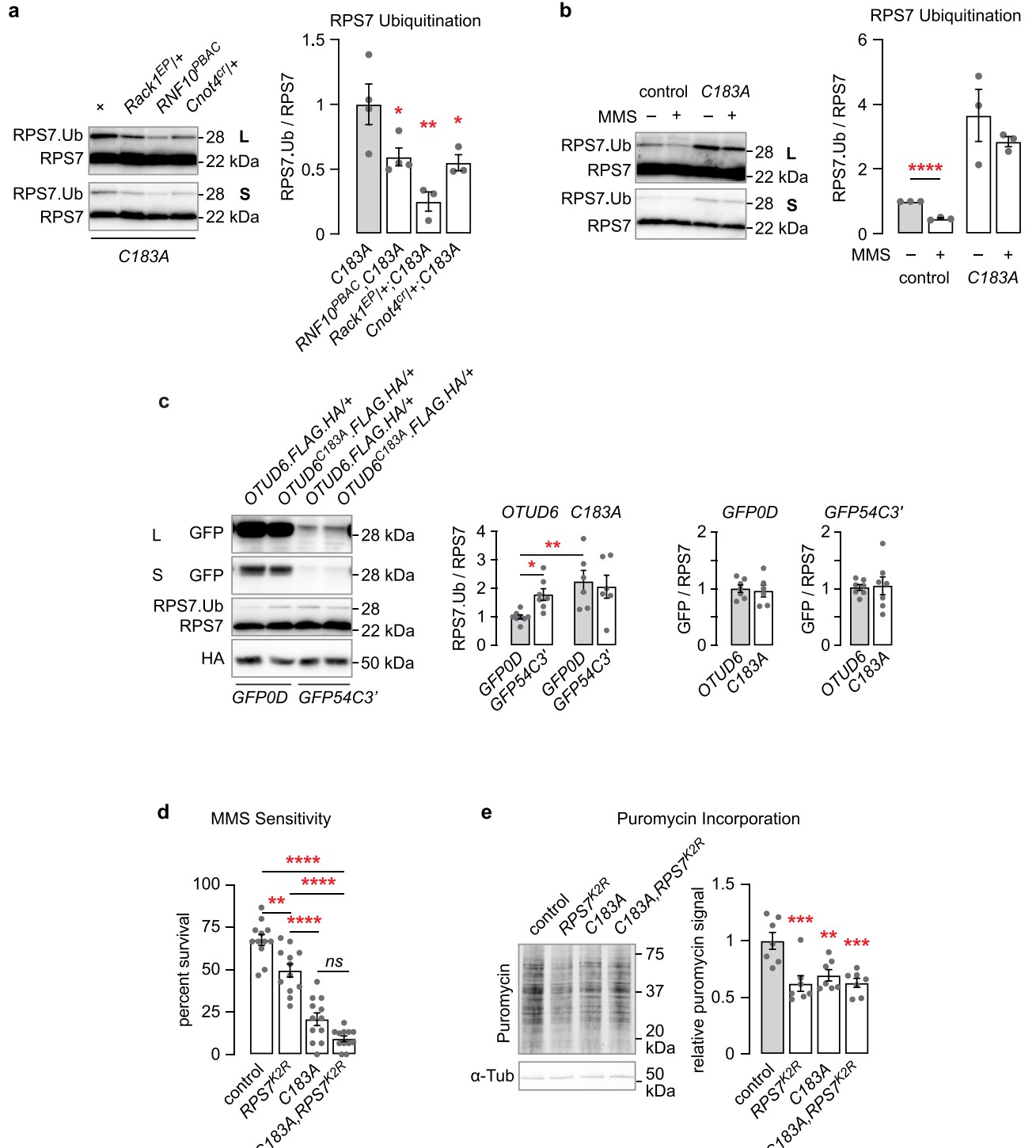

**Fig. 6 | RPS7 monoubiquitination is regulated by OTUD6 interacting proteins and by MMS. a** RPS7 monoubiquitination in head extracts from *OTUD6^C183A* alone and in combination with mutations in *Rack1*, *RNF10*, and *Cnot4*. ANOVA/Dunnett's, compared to *OTUD6^C183A*. (*n* = 4, 4, 3, 3). **b** RPS7 monoubiquitination is decreased by MMS treatment in head extracts. Two-tailed t-test compared to untreated controls. (*n* = 3, 3, 3, 3). **c** Levels of GFP, RPS7, and OTUD6.FLAG.HA in flies ubiquitously expressing GFP comprised of all common (*GFP0D*) and approximately 50% nonoptimal (*GFP54C3'*)

codon content. One way ANOVA with Sidak's multiple comparisons test. (Left: *n* = 8, 7, 6, 6. Middle: *n* = 6, 6. Right: *n* = 7, 7). **d** MMS sensitivity of *OTUD6^C183A*,*RPS7^K2R* flies. ANOVA/Sidak's, planned comparisons. (*n* = 12, 12, 12, 12). **e** Protein translation levels detected by puromycin incorporation in *OTUD6^C183A*,*RPS7^K2R* flies. ANOVA/Dunnett's, compared to control. (*n* = 7, 7, 7, 7) L: long exposure; S: short exposure. Data are presented as mean values +/− SEM. Dots on on bar graphs represent biological replicates. Source data and statistics are provided as a Source Data file.

wild-type or catalytically inactive OTUD6, detected via the translation initiation factor eIF2α[43]. In the IP-MS experiment with tagged OTUD6^C183A, we did not detect any ribosomal cofactors for the 43 S or 48 S preinitiation complexes, proteins implicated in ribosome quality

control pathways, or proteins implicated in ribosome biogenesis and maturation (Fig. 2a, Supplementary Data 1). Sucrose density gradient centrifugation and western blot analysis of gradient fractions confirmed that OTUD6^C183A associated with the free 40 S ribosomal subunit

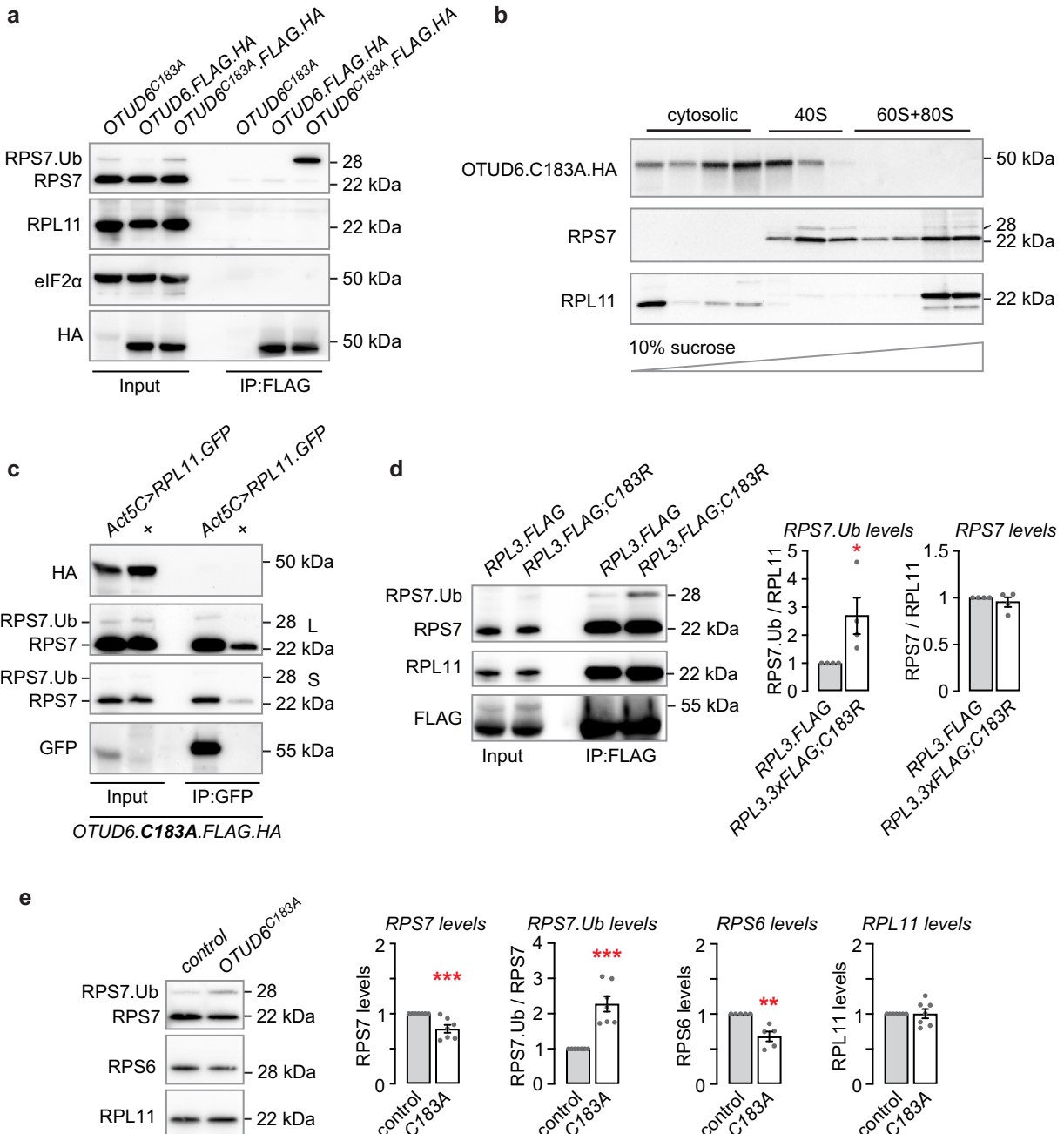

**Fig. 7 | OTUD6 associates specifically with the RPS7-monoubiquitinated free 40 S ribosomal subunit to regulate 40 S ribosome occupancy on mRNA. a** Co-IP in the presence of cycloheximide with tagged OTUD6 and OTUD6$^{C183A}$, or untagged OTUD6$^{C183A}$ as a control, probed for RPS7 (40 S), RPL11 (80 S), and eIF2α (43 S and 48 S preinitiation complexes). Blots are representative of 3 biological replicates. **b** Sucrose density fractionation and western detection of tagged OTUD6$^{C183A}$, RPS7, and RPL11. Blots are representative of 4 biological replicates. **c** Co-IP in the presence of cycloheximide with GFP-tagged RPL11 (*UAS-Rpl11.GFP*) expressed ubiquitously (*Act5C-Gal4*) in OTUD6$^{C183A}$.FLAG.HA flies, probed with HA, RPS7 and GFP. Blots are representative of 3 biological replicates. **d** Co-IP in the presence of cycloheximide

with endogenously FLAG-tagged RPL3 in control and *OTUD6$^{C183R}$*, probed for RPS7, RPL11, and FLAG. Quantification of RPS7 non-ubiquitinated and monoubiquitinated forms. Mann-Whitney test (two-tailed). (*n* = 4, 4, 4). **e** mRNA capture with oligo(dT) beads, probed by western analysis with antibodies for RPS7, RPS6, and RPL11. Ratio of *OTUD6$^{C183A}$* to genetic background control, except for RPS7 monoubiquitination. Mann-Whitney test (two-tailed). (*n* = 7, 7, 7, 7, 6, 6, 7). L: long exposure; S: short exposure. Data are presented as mean values +/− SEM. Dots on on bar graphs represent biological replicates. Source data and statistics are provided as a Source Data file.

(Fig. 7b, Supplementary Fig. 6A). Thus, in *Drosophila* OTUD6 appears to interact specifically with 40 S ribosomal subunits that contain monoubiquitinated RPS7.

Next, we asked if OTUD6 could be detected on immunoprecipitated 80 S ribosomes. To do this, we used GFP-tagged RPL11 (*UAS-RpL11.GFP*) that was expressed in all cells (*Act5C-Gal4*) in flies homozygous for either *OTUD6.FLAG.HA* or *OTUD6$^{C183A}$.FLAG.HA*. We were

unable to detect either wild-type OTUD6 or catalytically inactive OTUD6 on the 80 S (Fig. 7c, Supplementary Fig. 6B). The 80 S was intact because we recovered RPS7. Thus, we were unable to detect OTUD6 associated with the 60 S large ribosomal subunit or with 80 S ribosomes. Taken together, these data suggest that OTUD6 specifically interacts with monoubiquitinated RPS7 on the 40 S subunit that is not part of the translating ribosome.

Whereas OTUD6 was associated with RPS7.Ub on the free 40 S ribosomal subunit, RPS7 is also reported to be monoubiquitined during assembly of the 80 S at initiator methionines and when the 80 S encounters nonoptimal codons[7,28,30,31]. We asked if loss of OTUD6 deubiquitinase activity affected RPS7 monoubiquitination on the 80 S by immunoprecipitating the 80 S ribosome using RPL3 enhancer driven *RPL3.FLAG* transgene that is expressed ubiquitously, in the presence of cycloheximide. RPL3 immunoprecipitation revealed increased monoubiquitinated RPS7 in *OTUD6^{C183R}* vs. wild-type *OTUD6* (Fig. 7d). Thus, RPS7 monoubiquitination was increased on both the 80 S and the free 40 S.

Finally, to probe the impact of OTUD6 deubiquitinase activity on ribosomes on mRNA, we captured poly(A) mRNA under non-denaturing conditions using oligo-d(T) beads in wild-type and catalytically inactive OTUD6 flies and detected mRNA bound ribosomal proteins by western analysis. Probing for 40 S and 80 S markers revealed unchanged RPL11, decreased RPS6, decreased non-ubiquitinated RPS7, and increased monoubiquitinated RPS7 in *OTUD6^{C183A}* (Fig. 7e). The ribosomal subunits were mRNA bound because these interactions were markedly reduced with the addition of RNase prior to mRNA capture (Supplementary Fig. 6C). Thus, in flies with catalytically inactive OTUD6, the quantity of 80 S bound on mRNA appears unchanged, and the 43 S/48 S preinitiation complexes bound on mRNA were decreased. OTUD6 deubiquintation of RPS7 may thus promote incorporation of the 40 S into the 43 S and binding to mRNA.

### OTUD6 protein abundance and RPS7 monoubiquitination are regulated physiologically and by stressors

Our findings support OTUD6 deubiquitination of monoubiquitinated RPS7 on the free 40 S ribosomal subunit to regulate the level of protein translation. We asked if regulation of OTUD6 itself is a mechanism used by cells to control the level of protein translation. MMS treatment causes decreased RPS7 monoubiquitination, suggesting that OTUD6 abundance might be increased (Fig. 6b). MMS treatment did indeed increase OTUD6 protein abundance, suggesting that alkylation stress upregulates OTUD6 protein synthesis or stability (Fig. 8a). Protein translation is decreased as organisms age; this holds true in *Drosophila*[44]. We observed increased lifespan in OTUD6 mutants, that is consistent with decreased protein translation in the mutants (Fig. 3b). OTUD6 protein abundance was lower in old flies, suggesting that flies might tune down protein translation by decreasing OTUD6 levels as they age (Fig. 8b).

RPS7 monoubiquitination on the free 40 S ribosomal subunit may be due to complications encountered during protein translation, including ribosome stalling and the presence of nonoptimal codons in translating mRNAs[9,31]. Regulation of OTUD6 abundance or activity may also contribute to number of 40 S ribosomal subunits that are RPS7 monoubiquitinated. The pathogenic dipeptide repeat protein poly(GR) causes ribosome stalling and collisions on poly(GR) encoding mRNAs and general interruption of translation elongation[45,46]. Eye-specific expression (*GMR-Gal4*) of 36 repeat poly(GR) (*UAS-poly(GR)*) causes neurodegeneration of the eye (Fig. 8c)[47]. Catalytically inactive OTUD6, or OTUD6 RNAi driven in the same neurons as poly(GR), enhanced neurodegeneration, whereas OTUD6 overexpression marginally improved eye morphology, the latter rescuing bristle morphology towards wild-type (Fig. 8c, d). We next measured OTUD6.FLAG.HA protein levels and RPS7 monoubiquitination with poly(GR) expression. Critically, OTUD6 protein abundance was reduced and RPS7 monoubiquitination was increased, without apparent change in the abundance of non-ubiquitinated RPS7 (Fig. 8e). Thus, poly(GR) induced translation stress decreased OTUD6 abundance and increased RPS7 monoubiquitination. Taken together, these findings indicate that OTUD6 protein abundance can be regulated to affect the level of RPS7 monoubiquitination and global levels of protein translation.

## Discussion

*Drosophila* OTUD6 deubiquitinates monoubiquitinated RPS7 on the free 40 S subunit of the ribosome. The CNOT4 and RNF10 E3 ligases, along with the 40 S ribosomal protein RACK1, act upstream of OTUD6 to coordinate monoubiquitination of RPS7. OTUD6 deubiquitination of RPS7 increases cellular levels of protein translation. Physiological regulation of OTUD6 protein abundance is a mechanism for regulating the overall level of protein translation. This pathway regulates organismal sensitivity to alkylation and oxidative stress, and disruption of translation by dipeptide repeat proteins, and it may contribute to decreased protein translation in aging. RPS7 deubiquitination by OTUD6 may permit the free 40 S ribosomal subunit to progress to the 43 S preinitiation complex, providing a bidirectionally regulated metering mechanism for protein translation initiation.

OTUD6 specifically interacts with the free 40 S ribosomal subunit in *Drosophila*. Immunoprecipitation of catalytically inactive OTUD6 recovered 40 S but no 60 S subunits as detected by mass spectrometry, and by western analysis of OTUD6 wild-type and catalytically inactive immunoprecipitates in the presence of cycloheximide. Moreover, pull down of the 80 S ribosome recovered neither wild-type nor catalytically inactive OTUD6, and catalytically inactive OTUD6 co-fractionated with the free 40 S and not the 80 S on a sucrose density gradient. Hence, OTUD6 is not detectable on the 80 S. We were also unable to detect eIF2α in wild-type or catalytically inactive OTUD6 immunoprecipitates, suggesting that OTUD6 is not associated with the 43 S preinitiation complex nor with the 48 S initiation complex, or that it associates with a small fraction of these complexes in the cell. Finally, proteins that bind to the 40 S subunit during ribosome biogenesis or during ribosome recycling were absent from the proteins detected by mass-spectrometry from immunoprecipitation of catalytically inactive OTUD6. This is in contrast to a study done in yeast, where OTU2 was similarly associated with the 40 S but also with preinitiation complex factors, possibly reflecting changes in OTUD6/OTU2 protein function over evolutionary time[14]. Catalytically inactive OTUD6 may capture a precise step of the 40 S ribosome life cycle, a step where the free 40 S is not associated with other ribosome associated factors.

Catalytically inactive OTUD6 also captured the RNA exosome, that we verified as a bona fide interaction by western analysis. Mutations in the catalytic domain of the exosome, DIS3, suppressed *OTUD6^{C183A}* sensitivity to MMS. This may suggest that OTUD6 opposes RNA exosome dependent turnover of RNA in some capacity. However, we were unable to determine the nature of the interaction – whether it was direct with OTUD6, or indirect through the 40 S ribosome or another protein. The Ski complex that bridges the RNA exosome to the ribosome in ribosome biogenesis and recycling, was also not recovered[36,48].

Monoubiquitinated RPS7 is a major substrate for OTUD6 deubiquitinase activity. RPS7 was enriched in mass spectrometry of catalytically inactive OTUD6 immunoprecipitation. RPS7 was also enriched by mass spectrometry on the pool of ubiquitinated proteins isolated from catalytically inactive OTUD6. RPS7 monoubiquitination was increased in catalytically inactive OTUD6 flies, and it appeared to be the only detectable form of RPS7 that immunoprecipitated with tagged catalytically inactive OTUD6. Consistent with our findings, the yeast ortholog of OTUD6, OTU2, deubiquitinates yeast RPS7/eS7A on the 40 S ribosomal subunit[14,49,50]. The lysine on yeast RPS7/eS7A that is deubiquitinated by OTU2 is conserved in flies and it is deubiquitinated by OTUD6, and the lysine for ubiquitination is also conserved in human RPS7. While ubiquitination/deubiquitination of human RPS7 at the conserved lysine is not yet reported in mammals, OTUD6B was recovered on the human 40 S ribosomal subunit in kinase-dead RIOK1 cells[51]. RPS7 monoubiquitination was undetectable in both wild-type and catalytically inactive OTUD6 when the evolutionarily conserved monoubiquitination site was mutated in *Drosophila* RPS7. We were unable to detect wild-type OTUD6

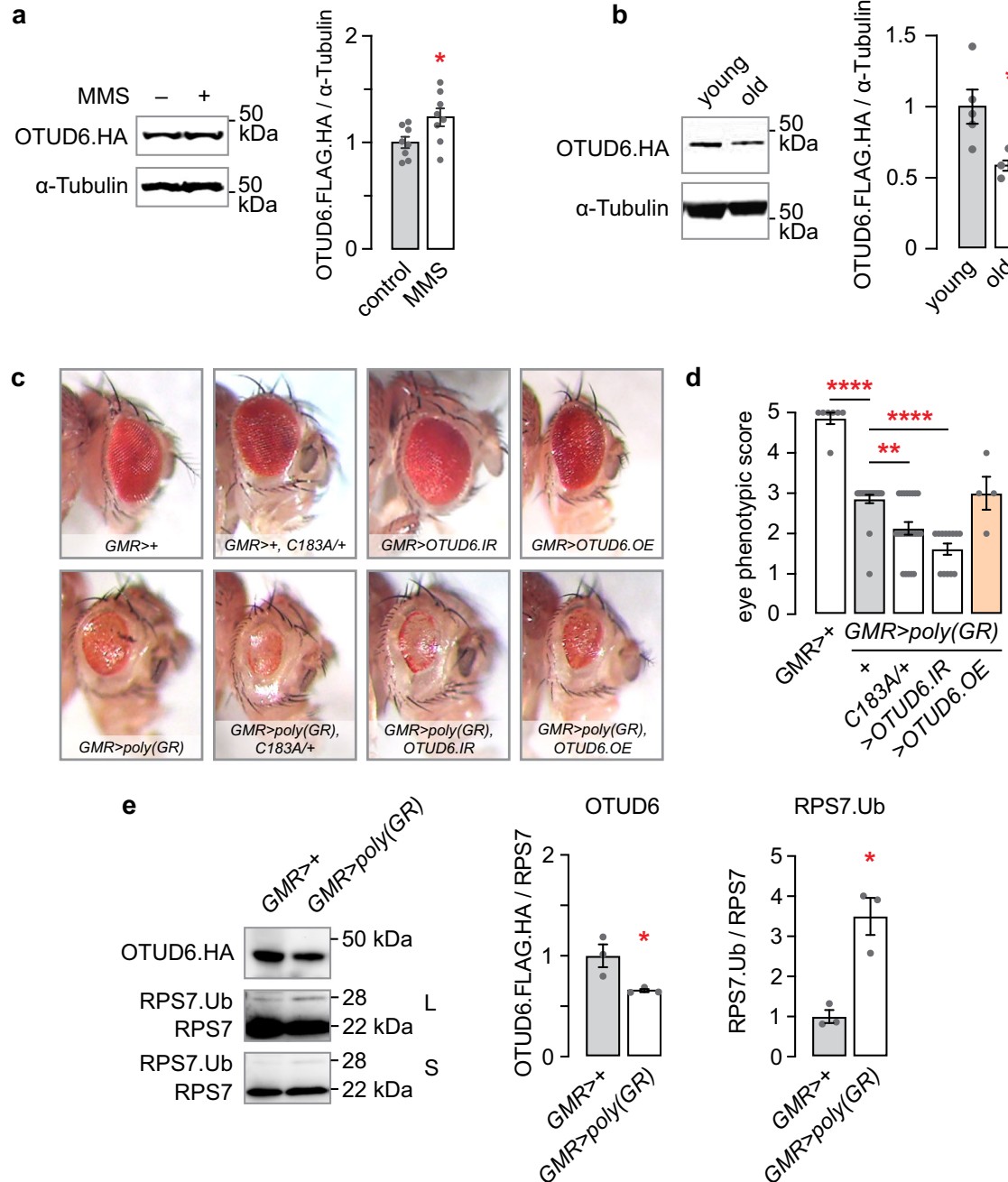

**Fig. 8 | OTUD6 protein levels are regulated by conditions that alter protein translation levels and efficacy. a** Western blot of OTUD6.FLAG.HA whole fly lysate probed for OTUD6 following 24 h 0.05% MMS exposure. Two-tailed t-test. (*n* = 8, 8). **b** OTUD6.FLAG.HA levels in young (3-5 days post eclosion) and old (30-35 days post eclosion) fly heads. Mann-Whitney test (two-tailed). (*n* = 5, 5). **c** Expression of poly(GR) repeat proteins in the *Drosophila* eye using *GMR-Gal4*, in *OTUD6^C183A^/+*, with coexpression of OTUD6 RNAi (*UAS-OTUD6.IR*) or OTUD6 overexpression (*UAS-OTUD6.OE*). Example eye phenotypes. **d** Quantification of the poly(GR) eye phenotypes. One way ANOVA with Sidak's multiple comparisons test. (*n* = 7, 21, 23, 13, 4). **e** Levels of OTUD6.FLAG.HA and RPS7 in head lysate of flies expressing poly(GR). OTUD6.FLAG.HA levels: Two-tailed t-test, RPS7 monoubiquitination: Mann-Whitney. L: long exposure; S: short exposure. Data are presented as mean values +/− SEM. (*n* = 4, 4, 4, 4). Dots on bar graphs represent biological replicates. Source data and statistics are provided as a Source Data file.

associated with RPS7 or with the ribosome, suggesting that OTUD6 association with the free 40 S ribosomal subunit is dependent on RPS7 ubiquitination status. Thus, a central and evolutionarily conserved function of OTUD6 is deubiquitination of monoubiquitinated RPS7 on the free 40 S ribosomal subunit. The RPS7 ubiquitination site is physically close to the preinitiation complex components eIF3c and eIF4A, such that monoubiquitinated RPS7 may change their interactions with the 40 S to influence the dynamics of pre-initiation complex formation or stability[14,52].

RPS7 monoubiquitination is proposed to arise from specific translation-related events that may feed into a pool of ubiquitin-modified free 40 S ribosomal subunits. RPS7 is reported to be ubiquitinated when the 48 S initiation complex transitions to forming the 80 S at sites of RNA translation initiation, following ribosome collisions, and when translating 80 S ribosomes pause at non-optimal codons[7,28,31]. At our current state of knowledge, this suggests that ribosome quality control mechanisms can split the ribosome while RPS7 is ubiquitinated from earlier steps of quality control. For

example, RPS7-monoubiquitinated 40 S subunits might be released when a nonoptimal codon is unable to be resolved. Additional, less completely characterized mechanisms that result in monoubiquitination of RPS7 exist, including in oxidative and endoplasmic reticulum stress[18,53]. OTUD6 may clean up 40 S subunits that acquire marks from specific translation events, to return them to competency for reinitiating protein translation.

OTUD6 protein expression levels can set the amount of protein translation in cells, and it may do so by metering the availability of initiation-competent 40 S ribosomal subunits via RPS7 deubiquitination. Critically, we found that OTUD6 overexpression increased and OTUD6 underexpression decreased protein translation, indicating that OTUD6 levels are rate limiting for a step in determining global protein translation rates. Delayed development and increased lifespan, but no other ribosomopathy-like phenotypes in OTUD6 loss-of-function and catalytically inactive mutants, is consistent with decreased protein translation[44]. Deletion of OTU2 in yeast also decreased translation, supporting a generally conserved function for OTUD6[49]. Importantly, physiological events can regulate the abundance of OTUD6 protein. OTUD6 protein levels declined with age, providing a potential mechanism for age-dependent decrease in protein translation levels; additional experiments are needed. Two additional conditions that affect the levels protein translation, alkylative stress and expression of poly(GR) dipeptide repeat protein, also altered OTUD6 protein abundance, suggesting that regulation of OTUD6 protein levels can regulate the levels of protein translation in a variety of cellular pathways[46,54,55]. RPS7 monoubiquitination was tightly coupled with OTUD6 protein abundance, suggesting that OTUD6 regulates the level of protein translation through deubiquitination of RPS7; increased protein translation is coupled to increased OTUD6 protein abundance and decreased RPS7 monoubiquitination. Finally, the decreased levels of 40 S (including 43 S/48 S initiation complexes) but not 80 S ribosomes associated with mRNA in catalytically inactive OTUD6 flies suggests that OTUD6 deubiquitination of RPS7 may control the amount of 43 S and 48 S complexes on mRNA. This provides a plausible mechanism for OTUD6 regulation of global protein translation, since translation initiation is the rate limiting step[56–58]. Regulation of OTUD6 deubiquitination of RPS7 appears to join a list of regulatory mechanisms for setting the rate of protein translation initiation[3]. A prediction is that OTUD6 ties the cellular levels of ribosome quality control to the cellular levels of protein translation. 40 S subunits also enter the translation initiation-competent pool through ribosome biogenesis, however currently there are no known roles for RPS7 monoubiquitination in biogenesis.

Regulation of RPS7 monoubiquitination is also involved in the response to alkylative stress. The stressor MMS increased OTUD6 protein abundance and decreased RPS7 monoubiquitination, changes that are predicted to increase global protein translation. MMS, however, dramatically downregulates global protein translation in a variety of organisms, including Drosophila, with a recent report tying this effect to activation of the integrated stress response pathway[42]. Upregulation of the OTUD6-RPS7 pathway may be a homeostatic response to moderate the impact of alkylation stress on global protein translation. Importantly, the MMS sensitivity of OTUD6 mutants allowed us to identify CNOT4, RNF10, and RACK1 as upstream promoters of RPS7 monoubiquitination and the regulation of alkylation stress. CNOT4/NOT4 is proposed to ubiquitinate RPS7 on the 48 S and on the 80 S ribosome[7,28,30,31]. RNF10 monoubiquitinates RPS2 and RPS3 in iRQC, a form of ribosome quality control during translation initiation[25]. RNF10 also ubiquitinates the same two subunits in 80 S disome collisions downstream of ZNF598 ubiquitination of RPS10, and in a second, collision- and ZNF598-independent role tied to slowed translation elongation[33,59]. It is not known if RPS7 is also a substrate during these translation events, however we note that we did not uncover an interaction between ZNF598 and OTUD6. RACK1

coordinates specific ubiquitination steps and ribosomal quality control more broadly, through mechanisms that are partly understood[22,60–62]. These proteins, and RNF123 that also interacts with OTUD6, reveals that additional pathways exist that regulate RPS7 ubiquitination.

## Methods

### Drosophila husbandry and strains
All flies were reared on standard cornmeal molasses food. All Drosophila strains were outcrossed five or more times to the $w^{1118}$ Berlin genetic background (Control). Flies were reared at 25 °C at 60% humidity on a 12:12 light:dark cycle. Supplementary Data 4 lists Drosophila strains, antibodies and conditions, recombinant DNA, oligonucleotides, chemicals, software, and their sources.

### Generation of OTUD6 and RPS7 mutants
To generate UAS-OTUD6.FLAG.HA, a plasmid containing FLAG.HA-tagged OTUD6 under control of UAS enhancer elements was obtained from the Drosophila Genome Resource Center (UFO07783). The plasmid was injected into flies containing the VK00018 attP integration site (Bloomington Drosophila Stock Center (BDSC) 9736), by BestGene (Chino Hills, CA). CRISPR mutagenesis was used to alter the sequences of endogenous OTUD6 and RPS7. To generate epitope-tagged wild-type and catalytically inactive OTUD6 mutants and RPS7 mutants, a guide RNA (gRNA) was created that had homology to sequences near the site of mutagenesis. A single stranded oligodeoxynucleotide (ssODN) containing C183A, C183R, or no mutation, with or without an added FLAG.HA tag was used as a template for integration into the OTUD6 locus. An ssODN containing K83R,K84R in RPS7 was used a template for integration into the RPS7 locus. All guide RNA constructs were cloned into the pU6-BbsI-chiRNA plasmid and injected into yw; attP40{nos-Cas9}/CyO flies by BestGene.

### Paraquat and MMS exposure
Flies were collected 2–5 days post-eclosion and groups of fifteen males were fed 10 mM paraquat (Sigma 856177), 0.05% methyl methanesulfonate (MMS) (Fisher scientific AC156890050), or vehicle as control, on low melt agarose containing 5% sucrose. For paraquat, survival was assayed every 12 h over 72 h. For MMS, survival was assayed every 2 hours from 24 to 36 h. Each exposure was repeated across a minimum of 3 separate days for a minimum of 12 biological replicates of 15 males per genotype tested. Experimenter was blinded to genotype.

### X-ray irradiation
A 6-hour timed collection was performed at 25 °C using Drosophila apple juice agar plates[63]. Twenty 3rd instar larvae were moved to dishes containing normal food. The larvae were exposed to 110 rad/min X-ray irradiation until desired dose. Relative survival was determined by the number of larvae that survived to adulthood. Experimenter was blinded to genotype.

### DR-white DNA damage reporter
To induce double strand breaks, virgin females containing DR-white and WT or mutant OTUD6 were crossed to males containing the heat-inducible I-SceI transgene in addition to WT or mutant OTUD6. After 3 days, the adults were removed, and embryos were heat shocked in a 37 °C water bath for 1 h. Single male progeny from these crosses were then crossed to 2-4 yw virgins. Progeny from 15 to 22 individual male crosses were phenotypically scored for DNA repair events[64].

### Olfactory classical conditioning behavioral analysis
Olfactory aversive learning was assayed using previously described methods (Tully and Quinn, 1985; Stahl et al. 2022). Briefly, 2–6 day old flies were conditioned under dim red light at -22 °C. 60 flies with 50:50 male:female ratio were trained using two odors: ethyl butyrate and

isoamyl acetate. Each odor was diluted to 0.06–0.1% v/v in mineral oil, empirically adjusted to achieve a 50:50 naïve distribution when presented to flies in a T-maze. Odors were presented by bubbling in a scintillation vial. For conditioning, each set of flies was presented one odor (the CS + ) paired with an electric shock aversive unconditioned stimulus (US) (12x pulses of 90 V) for 1 min, followed by 30 s of air, and the second odor (the CS-) presented with no US for 1 minute. Following conditioning, flies were then placed into a T-maze for choice discrimination between the CS+ and CS- containing arms for 2 min. Control experiments testing odor and shock avoidance allowed the flies to choose arms containing the odor or mineral oil (odor avoidance) or contained a set a shock grids with one arm activated every 5 s at 90 V (shock avoidance) and the alternative arm was inactive. A performance index (PI) was calculated as the number of flies choosing the CS- arm minus flies in the CS+ arm/(CS- + CS + ). Avoidance was calculated as flies choosing the arm with odor/shock minus flies in arm absence the stimulus/total number of flies in the two arms. Statistical analysis utilized ANOVA (one-way parametric). Multiple comparisons within genotypes were analyzed with Sidak's post hoc test with α=0.05. Experimenter was blinded to genotype.

## Eclosion timing
To quantify developmental timing a timed collection was performed for 6 h at 25 °C. Following the collection 50 eggs per replicate were moved to vials containing normal food and kept at 25 °C. Number of flies to eclose was counted daily at the same time each day. Survival from egg to adulthood was determined by taking percentage of the number of eggs that survived to adult hood vs the total number of eggs seeded into the vials. Experimenter was blinded to genotype.

## Western analysis
5–10 adult fly heads were homogenized in RIPA buffer (Sigma 20-188) that was supplemented with 1x Complete, Mini, EDTA-free protease inhibitor. The homogenate was centrifuged at 12,000 x g for 5 minutes at 4 °C. 30 μL of the lysate was retained and recombined in 10 μL of 4x NuPage LDS buffer (NP0007) containing 5% 2-mercaptoethanol (Sigma M6250). The samples were heated for either 3 minutes at 95 °C or 10 min at 70 °C. Protein levels were quantified using Pierce™ BCA Protein Assay Kit and 20–40 μg of lysate was separated on an Invitrogen 4–12% Bis-Tris gel (NP0322) using MOPS running buffer (NP0001). Proteins were transferred onto a PVDF membrane (Millipore IPFL00010), blocked in either 5% non-fat dry milk or 5% BSA and incubated in primary antibody (Supplementary Data 4) overnight at 4 °C. Blots were imaged using a Li-Cor Odyssey or BioRad ChemiDoc Imaging System and analyzed using Fiji.

## Sucrose density fractionation
Five hundred *Drosophila* heads were homogenized in 1 mL of PBS supplemented with 100 μg/mL cycloheximide and 1% Triton X-100. Following a 15 min incubation on ice, lysates were centrifuged at 20,000 x g for 20 min at 4 °C. Supernatant was layered onto a 10–45% linear sucrose gradient (40 mM Tris-HCl pH 7.4, 150 mM NaCl, 5 mM MgCl$_2$, 100 μg/mL cycloheximide) and centrifuged in SW40ti swinging bucket rotor at 155,000 x g for 3 h at 4 °C. Following centrifugation, 0.5 mL fractions were collected using a BioComp piston gradient fractionator. The fractions were precipitated with trichloroacetic acid (TCA) and washed twice with acetone and examined by western.

## Quantification of global protein translation
To assess protein translation levels in the brain, 2–4 day old flies were starved for 6 hours before being fed 600 μM puromycin in a 5% sucrose solution containing 3% ethanol for 16-18 hours[65]. Flies were decapitated and homogenized in RIPA buffer, and 20 μg of lysate was used for western blotting. Puromycin incorporation was detected using an antibody to puromycin and normalized to α-Tubulin.

## Antibody coupling to beads and co-immunoprecipitation
To make anti-FLAG beads, 10 μg of mouse anti-FLAG (F3165; Sigma-Aldrich) antibody was bound per 1 mg of beads Dynabeads M-270 epoxy (143-02D; Invitrogen)[66]. Antibody was coupled to the beads using the coupling kit protocol and the beads were suspended at 10 mg/mL. To perform co-IP, 900x tagged OTUD6$^{CI83A}$ and untagged OTUD6$^{CI83A}$ flies, control, were collected. Flies were decapitated using liquid nitrogen and collected using a sieve. The heads were homogenized in 1 mL of lysis buffer (10 mM HEPES buffer PH7.4, 150 mM NaCl, 5 mM MgCl$_2$, and 0.5% Triton X-100). The lysate was centrifugation at 20,000 x g for 20 min at 4 °C and filtered through a 0.45 μm low protein binding PVDF syringe filter (Millipore, SLHVX13NL). 500 μL of extract was incubated with 200 μl of prewashed mouse anti-FLAG magnetic beads at 4 °C for 30 min. The beads were washed four times in wash buffer (10 mM HEPES buffer PH7.4, 150 mM NaCl, 5 mM MgCl$_2$, and 0.1% Triton X-100) and two additional washes without detergent and protease inhibitor. For co-immunoprecipitation with OTUD6$^{CI83A}$.FLAG.HA versus OTUD6$^{CI83A}$, the beads were eluted in 30 μL of 500 ng/μL 3xFLAG peptide (F4799, Sigma) at 4 °C for 30 min with mixing. For serial monoubiquitin enrichment in OTUD6$^{CI83A}$.FLAG.HA, 100 μg/ml Cycloheximide and 100 mM iodoacetamide was added to the lysis and wash buffer. Additionally, the beads were eluted in 30 μL 1x LDS buffer at 95 °C for 5 min.

## Serial monoubiquitin enrichment
To couple ubiquitin binding domains, purified recombinant protein was coupled to prewashed NiNTA agarose beads (30210, Qiagen) at 1 mg/mL for 1 hour at 4 °C with rotation. The beads were washed three times with wash buffer (0.1% NP-40, 150 mM NaCl, and 50 mM Tris pH 7.5) prior to use. Approximately 400 flies were collected, and head extracted. The heads were homogenized in 1 mL of denaturing lysis buffer (50 mM Tris-HCL pH 7.5, 150 mM NaCl, 1 mM EDTA, 1% NP-40 and 10% glycerol) supplemented with protease inhibitor tablet and 50 mM N-ethylmaleimide. The lysate was centrifuged at 20,000 xg at 4 °C for 20 min and filtered through a 0.45 μm low protein binding PVDF syringe filter. 3 mg of protein in a final volume of 500 uL was added to 50 μL 1:1 slurry containing His-HALO-hPlic1UBA4x. The bead lysate mix was incubated for 1 h at 4 °C with rotation. The protein extract was removed from the beads and 400 μL was added to prewashed beads containing His-hP2-UBA and incubated at for an additional 1 h at 4 °C with rotation. The beads were washed 4x times with 1 mL wash buffer (50 mM Tris, 0.1% NP-40, and 150 mM NaCl). The proteins were either digested on the beads for MS or eluted in 30 μL of 1x loading buffer at 95 °C for western blotting.

To capture all ubiquitin, 50 whole flies were homogenized in 1 mL of denaturing lysis buffer (50 mM Tris-HCL pH 7.5, 150 mM NaCl, 1 mM EDTA, 1% NP-40 and 10% glycerol) supplemented with protease inhibitor. The lysate was centrifuged at 20,000 x g at 4 °C for 20 min and filtered through a 0.45 μm low protein binding PVDF syringe filter. 500 μL of lysate was loaded onto prewashed ChromoTek Ubiquitin-Trap Magnetic Agarose beads and was incubated for 1 h at 4 °C with rotation. The beads were washed with 1 mL of denaturing lysis buffer 3x times before being eluted in 30 μl of 1x loading buffer at 95 °C.

## Mass Spectrometry analysis of Co-Immunoprecipitated Proteins
**In-gel digestion.** Eluted samples were run with 4-12% Bis-Tris SDS-PAGE gel (Novex, Invitrogen) at 125 V for 15 min and separated approximately 1.5 cm using MOPS buffer. The gel was stained with Coomassie, and the entire mobility region was excised into small gel pieces. The in gel digestion was performed as following: (i) gel pieces were washed with 1x ddH$_2$O followed by dehydration with 50% acetonitrile/50% H$_2$O solution at room temperature for 15 min; (ii) gel

pieces were dried in the speed vac for 10 min; (iii) trypsin digestion was performed in 50 mM ammonium bicarbonate, pH 8.0; (iv) trypsin digestion reaction was performed at 37 °C for overnight; (v) the reaction was quenched with 1 μL of 1% acetic acid before peptide extraction with 50% acetonitrile/45% ddH$_2$O/5% formic acid at room temperature for 15 min.

**LC-MS and Data Analysis.** The peptides were separated with Thermo UltiMate 3000 UHPLC system interfaced to a ThermoFisher Fusion Lumos. Peptides were loaded on a trapping column and eluted over a 75 μm analytical column at 400 nL/min. A 125 min gradient was employed for peptide separation. The mass spectrometer was operated in data-dependent mode with FAIMS source ion, with MS and MS/MS performed in the Orbitrap at resolution of 240000. APD was turned on. The instrument was run with 3 s cycle for MS and MS/MS. Collected data for all of the 3 independent biological replicates were searched with following parameters: (i) enzyme: Trypsin; (ii) database: Uniprot *Drosophila* (concatenated forward and reverse plus common contaminants); (iii) fixed modification: carbamidomethyl (C); (iv) variable modifications: Oxidation (M); mass values: monoisotopic; (v) precursor mass tolerance: 25 ppm; (vi) fragment ion tolerance: 0.8 Da; (Vii) allowed missed-cleavages: 2. Data were further filtered at 2% FDR at peptide and protein level. SAINT (Significance Analysis of INTeractome) analysis (Choi et al., 2011) was performed on the filtered dataset to find protein-protein interaction. The analysis used peptide spectral match (PSM) scoring function to assign a probabilistic scoring to find out the potential protein-protein interacting partners based on the log Odd score values. The positive log Odd score values indicate higher confidence in the prey protein to the bait.

## Mass Spectrometry analysis of serial poly- and mono-ubiquitination enrichments

**On bead digestion.** The beads were washed with 1 mL of ice-cold PBS and resuspended in 200 μL of 50 mM HEPES, pH 8.0. Proteins were reduced by incubation with 3 μL of 500 mM DTT at 37 °C for 45 min and 11.8 μL of 380 mM iodoacetamide at RT in dark for 30 min. To quench the overall reduction/alkylation reaction, an additional 3 μL of 500 mM of dithiothreitol was added for a period of 15 min in the dark at RT. Then, 100 ng of trypsin was added to the beads for 1 hr on a shaker at 37 °C. The supernatant was removed, an additional 100 ng of trypsin was added, and the digestion was performed overnight on a shaker at 37 °C. The digested samples were acidified with trifluoroacetic acid (2% final) and cleaned with C18 stage tips (Pierce, 87782). Peptide quantification was then performed using BCA colorimetric peptide kit (Pierce, 23275) and 20 μg of peptide per sample was used for TMT-16 plex labeling.

**TMT-16plex Labeling and High-pH Fractionation.** The dried peptides were resuspended in 100 μL of 200 mM HEPES, pH 8.0 buffer. The TMT reagent was solubilized in 20 μL of anhydrous acetonitrile and mixed with the peptide solution. The reactions were quenched with 5 μL 5% hydroxylamine (Sigma-Aldrich, 438227) 15 min at RT and the samples were pooled together. The pooled sample was desalted with Sep-Pak C18 column (Waters, WAT054955) and dried in a speed vac. The dried labeled peptide mixture was fractionated into 24 fractions using Thermo spin column kit (Pierce, 84868) and combined into final 12 fractions. The combined fractions were cleaned up with C18 spin tips (Pierce, 84850) and dried in speed vac prior to LC-MS analysis.

**LC-MS and data analysis.** A Dionex Ultimate 3000 RSLCnano system (Thermo Fisher) and an Orbitrap Eclipse Tribrid MS (Thermo Fisher) were used for analysis of the samples. Peptide separation was performed on 25 cm length and 75 μM diameter AURORA series column packed with 1.6 μM C18 material with pore size of 120 Å (Ion Optics,

IO2575011997). A linear LC gradient of 185 min with 2% to 30% buffer B (98% acetonitrile/0.1% formic acid) in buffer A (2% acetonitrile/0.1% formic acid) at flow rate of 300 nL/min. The sample analysis was performed using a multinotch MS3-TMT method and data dependent mode.

The data was searched using comet against a *Drosophila* database that included Uniprot *Drosophila* protein sequences[67]. Peptide and protein level data was passed through 2% false discovery rate separately following the previously published algorithm[68,69]. The searched dataset was further processed for TMT reporter ion intensity-based quantitation using the Mojave algorithm with an isolation width of 0.5[70].

**Quantification and statistical analysis.** Peptide spectral matches were filtered out from peptides with length less than 5; with isolation specificity less than 50%; with reporter ion intensity less than $2^8$ noise estimate; from peptides shared by more than one protein; with summed reporter ion intensity (across all 12 channels) lower than 30,000. Quantification and statistical analysis were performed by MSstatsTMT v2.2.7, an open-source R/Bioconductor package[71]. Multiple fractions from the same TMT mixture were combined in MSstatsTMT. To test the two-sided null hypothesis of no changes in abundance, the model-based test statistics were compared with the Student t-test distribution with the degrees of freedom appropriate for each protein and each dataset. The resulting P values were adjusted to control FDR with the method by Benjamini-Hochberg.

MSstatsTMT generated a normalized quantification report across all the samples at the protein level from the processed PSM report. Global median normalization equalized the median of the reporter ion intensities across all the channels and TMT mixtures, to reduce the systematic bias between channels. The normalized reporter ion intensities of all the peptide ions mapped to a protein were summarized into a single protein level intensity in each channel and TMT mixture. MSstatsTMT performed differential abundance analysis for the normalized protein intensities. MSstatsTMT estimated log2 (fold change) and the standard error by linear mixed effect model for each protein. The inference procedure was adjusted by applying an empirical Bayes shrinkage.

## RPS7 antibody generation

To create and RPS7 antibody we identified a region of antigenic sequence in the RPS7 peptide using and online implementation (http://imed.med.ucm.es/Tools/antigenic.pl) of a published method[72]. The peptide was synthesized by Sigma-Aldrich and the antibody raised in chickens by Pocono Rabbit Farm (PA). The final bleed was affinity purified to the peptide.

## Oligo(dT) mRNA Capture

400 fly heads were lysed in 1 mL of oligo(dt) lysis buffer (20 mM HEPES, pH 8, 125 mM KCl, 4 mM MgCl$_2$ and 0.05% NP-40) supplemented with Complete EDTA-free protease inhibitor. The extracts were centrifuged at 20,000 x g at 4 °C for 20 min and filtered through a 0.45 μm low protein binding PVDF syringe filter. 500 μL of the extract was loaded on to 200 μL of oligo(dT) beads equilibrated to 1 mL of oligo(dT) lysis buffer. The beads/extract mix was incubated for 30 min at 4 °C with rotation. The beads were washed 4x times with 1 mL lysis buffer and eluted in 30 μL of 1x loading buffer at 95 °C. For RNase treatment, a single extract was used and evenly split across oligo(dT) beads. Following the binding of the mRNA to the oligo(dT) beads, the flow through was removed and the beads were resuspended in 100 μL of 1x RNase H reaction buffer. 2 μL of RNase H or water was added to the respective tubes and the reactions were moved to room temperature for 15 minutes. The beads were then washed 4x times with 1 mL lysis buffer and eluted in 30 μL of 1x loading buffer at 95 °C.

## Immunohistochemistry

Adult female *Drosophila* were fed on standard culturing medium supplemented with 10% yeast paste to stimulate egg development. Adult fly brains and ovaries were dissected in PBS with 0.05% Triton X-100 (PBT), fixed overnight at 4 °C (brains) or 40 min at room temperature (ovaries) in PBT with 2% paraformaldehyde, blocked in PBS with 0.5% Triton X-100 with 5% normal goat serum and 0.5% bovine serum albumin (HDB), and immunostained as described previously[73]. Antibodies were rabbit anti-HA (1:300, Cell Signaling) and mouse anti-Bruchpilot (nc82, 1:20, Developmental Studies Hybridoma Bank, Iowa). Samples were mounted in Vectashield with DAPI (Vector Laboratories) and imaged on a Zeiss LSM-880 confocal microscope. Image stacks were processed in Fiji, and brightness and contrast were adjusted in Photoshop CC 2022 (Adobe).

## Statistics and reproducibility

Statistical tests and results are presented in the Source Data file. Statistical analysis was carried out in Prism v9.4.1. Two-tailed t-tests or Mann-Whitney pairwise tests (for data with unequal variance) were done for pairwise comparisons. One-way ANOVA followed by Tukey's post-hoc comparisons were used for multiple comparisons. If the data showed unequal variance by the Brown-Forsythe test, Welch's ANOVA followed by Dunnett's T3 post-hoc was used. If the data did not pass the Shapiro-Wilk normality of the residuals test, Kruskal-Wallis test followed with Dunn's post-hoc was used. Error bars are SEM. Dots overlaid on the bar graphs are the value measured for biological replicates. No statistical methods were used to predetermine sample sizes. Sample sizes are similar to those used in other *Drosophila* experimental paradigms. No data was excluded using statistical methods. For all experiments where flies were observed the experimenters were blinded to genotype or condition. All other experiments were not blinded.

## Reporting summary

Further information on research design is available in the Nature Portfolio Reporting Summary linked to this article.

## Data availability

All data needed to evaluate the conclusions in the paper are present in the Source Data file and in the Supplementary Information and Data files. Raw data from mass spectrometry were deposited within the MassIVE repository under the identifier MSV000091040. *Drosophila* strains created in this study are deposited at the Bloomington *Drosophila* Stock Center. Source data are provided with this paper.

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

## Acknowledgements

We thank Neha Kachewar for help with dual drug treatment experiments and genotyping for RPS7 CRISPR editing. Genentech Contract (FWW), R01NS124716 and R01NS114403 (SMT).

## Author contributions

Conceptualization: S.V., V.M.D., F.W.W. Methodology: S.V., P.D., A.S., T.H., C.M.R., D.S.K., S.M.T., F.W.W. Investigation: S.V., P.D., A.S., T.H.,

F.W.W. Visualization: S.V., P.D., A.S., F.W.W. Supervision: C.M.R., D.S.K., V.M.D., S.M.T., F.W.W. Writing—original draft: S.V., F.W.W. Writing—review & editing: S.V., P.D., C.M.R., D.S.K., F.W.W.

## Competing interests

The authors declare no competing interests.
