## [Transparent Peer Review file · Nature Communications]

OTUD6 deubiquitination of RPS7/eS7 on the free 40S ribosome regulates global protein translation and stress

Corresponding Author: Professor Fred Wolf

Version 0:

Reviewer comments:

Reviewer #1

(Remarks to the Author)

The manuscript by Villa and collaborators describes the identification of the OTUD6 deubiquitinase as an enzyme that deubiquitinates the ribosomal RPS7 protein on free 40S ribosomes and thereby promotes translation in *Drosophila*. They identify RACK1, CNOT4 and RNF10 as working upstream of OTUD6. They additionally suggest that the levels of the OTUD6 protein are limiting for this function and regulated in response to specific stresses (alkylation and translational stress) and upon ageing, to regulate translation. They suggest that the step that is affected is the rate limiting translation initiation step, by regulation of the amount of 40S ribosomes that are able to be recycled.

General:

This work is an interesting study that confirms and extends findings in budding yeast regarding the Otu2 ortholog of OTUD6, and its relevance for regulation of translation re-initiation. It is highly relevant and timely, and will be of general interest.

Major Comments:

- 1) For Figure 6B, the IP of RPL11-GFP does not reveal detectable ubiquitinated RPS7 in the IP, so it is not a good control to suggest that OTUD6 does not associate with 80S because OTUD6 is absent in the IP (see comment 2 where authors now show that ubiquitinated RPS7 is present in 80S)
- 2) Why do the authors argue that the increase of ubiquitination of RPS7 in the 80S in the OTUD6 C183R mutant suggests that the ubiquitination of RPS7 prevents RPS7-Ubi incorporation into the translating ribosome (Figure 6C)? Does this result per se actually argue for this?
- 3) If I understand correctly, the authors propose that the presence of one transgene can alter the level of RPS7 ubiquitination dependent upon the transgene content of non-optimal codons. Can the authors justify this interpretation? Is the transgene so much overexpressed that it can have an effect visible above all the ongoing translation in the cell. Also, the effect shown on Figure 6D is not convincing.
- 4) What is the evidence mentioned in the discussion that there are other routes to RPS7 deubiquitination according to the known routes for mono-ubiquitination? This is not clear.
- 5) If there is less re-initiation of 40S, and less 40S associated with the mRNA, should not also then less 60S rejoin the mRNA and then be less 60S?
- 6) The authors use many times RPS7-Ubi relative to overall Ubi for quantification, and this is really a very strange quantification. It should be as in Figure 7E, relative to RPS7 levels. Whether changes are then similar or not to overall ubiquitination level changes should be a separate measurement. I am also curious about how overall ubiquitination is evaluated. I could not find the description in the methods. As it stands, I am not convinced by these ratios (Figure 4D, 5A and 5B)

Minor comments

- 1) Do the authors suggest that their higher affinity OTUD6 C183R is dominant negative since it is MMS sensitive in the heterozygote?
- 2) The authors mention that no poly-Ubi RPS7 is detectable in the OTUD6 mutants but they do not show entire blots, this should be made available in supplementary data.
- 3) The staining in Figure 1G expressed to be in nucleus and nucleolus should be indicated (maybe arrow for an example of this), because this is not visible on the figure.
- 4) On Figure 4E, panel E, what are L and S (it seems high and low exposure, but there is no legend for this). Same comment for Figure S3D.
- 5) Figure 7A (and others): are the dots multiple experiments (biological repeats) or multiple repeats of the same samples (a

question that is generally relevant for the quantifications).

6) The ubiquitination of CNOT4 has never actually been shown to occur during 80S assembly during initiation or when non-optimal codons are encountered, nor has it been shown that this is coupled to mRNA deadenylation. These are all interpretations of observed data and should be presented as such (it has been proposed.....).

Reviewer #2

(Remarks to the Author)

In this manuscript, the authors examined the biological role of OTUD6 in *Drosophila*, which is one of the deubiquitinases. The authors identified the 40S ribosome subunits as OTUD6-interacting proteins by mass spectrometry and revealed that loss of the OTUD6 deubiquitylating activity led to the accumulation of mono-ubiquitylated RPS7. Protein synthesis was reduced in the OTUD6 mutants, whereas overexpression of OTUD6 enhanced protein translation with the decreased level of RPS7 ubiquitylation. Further, the protein synthesis defect in the OTUD6 mutant was suppressed with the mutations of some E3 ubiquitin ligases. The authors also purified mRNA-bound ribosomes and showed that 40S or 43S/48S bound on mRNA were decreased.

From these results, the authors suggest that deubiquitylation of RPS7 by OTUD6 is required for efficient protein synthesis and that OTUD6 controls the amount of 43S and 48S complexes on mRNA. However, it is not clear at what point OTUD6 removes ubiquitin from RPS7 during the ribosome cycle and how deubiquitylation of RPS7 mediates association of mRNAs with the ribosome. Further, it also remains to be elucidated whether the sensitivity of the OTUD6 mutant to MMS treatment results from impairment in deubiquitylation of RPS7, because MMS treatment reduced RPS7 monoubiquitylation in both wild-type and OTUD6 mutant flies. This reviewer would like to know whether MMS treatment accelerates protein translation and ameliorates the defect in protein synthesis in the wild-type and OTUD6 mutant flies, respectively. The authors did not examine the effect of MMS treatment on protein translation at all in this manuscript. Other minor points are also listed below.

1. There is no data to speculate how the mutations in DIS3 suppress MMS sensitivity in the OTUD6 mutant. Is it possible that the mutations in DIS3 reduce RPS7 monoubiquitylation?
2. It would be better to describe clearly which E2s and E3s are able to suppress the phenotype in the OTUD6 mutant in Figure 2D and supplementary figure 2.
3. At lines 149 and 150, the authors claimed that they did not detect bands that would be consistent with polyubiquitination of RPS7. However, this reviewer cannot determine if the authors' claim is correct with the current Fig. 4C because upper areas of the membranes were cut off.
4. Have the authors examined whether overexpression of RNF10 and RACK1 increase in monoubiquitylation of RPS7 and decrease in protein synthesis rates?
5. It is overstating to describe that "flies may tune down protein translation by decreasing OTUD6 as they age" at lines 229 and 230. The authors should examine whether overexpression of OTUD6 recovers protein translation even in aged flies.
6. In Figure 6B, this reviewer does not understand why monoubiquitinated RPS7 was not detected because RPS7 in this figure was purified from 80S ribosomes.
7. At lines 224 and 225, the authors supposed that increase in OTUD6 abundance led to decrease in monoubiquitylated RPS7 under MMS treatment. However, decrease in monoubiquitylated RPS7 occurred in the OTUD6 C183A mutant, implying that decrease in monoubiquitylated RPS7 is independent of OTUD6. The authors also should examine whether MMS treatment affects global protein translation rate or not in Figure 7A.

Reviewer #3

(Remarks to the Author)

The modifications of ribosomal proteins are critical for regulating translation activity. In this study, the authors investigate the role of OTUD6, a deubiquitinase, in specifically deubiquitinating the RPS7 subunit on the free 40S ribosome. The study demonstrates that OTUD6 interacts with RPS7 in the free 40S form, but not in 43S/48S initiation complexes or the translating ribosome. OTUD6 activity promotes the loading of free 40S on mRNA, thereby regulating protein translation. The abundance of RPS7 monoubiquitination and OTUD6 is responsive to translational and alkylation stress. The findings suggest that OTUD6 plays a crucial role in enhancing translation initiation by regulating the recycling of 40S ribosomes. Overall, the study provides insights into the regulation of ribosomes through ubiquitination/deubiquitination events and highlights the important role of OTUD6 in this process. The findings have implications for understanding protein translation regulation and cellular stress response. The study's discovery is novel, and the genetic analyses are abundant. However, some biochemical mechanisms remain inadequately illustrated. Major revision is recommended. Some specific comments are:

1. To definitively demonstrate the role of RPS7 ubiquitination, it would be important to identify the precise site(s) of ubiquitination and perform mutational analysis by substituting the Lysine residue(s) with Arginine. Then it would be possible to monitor the translational activity upon the mutation or analyze the growth under stress conditions. Some known human RPS27 ubiquitination sites may be found in the Phosphosite database (<https://www.phosphosite.org/proteinAction.action?id=6553&showAllSites=true>)
2. Is the ubiquitination regulation of RPS7 by OTUD6 conserved in mammalian cells? Considering that OTUD6B mutations are associated with human intellectual disability, it is highly relevant to show or at least discuss whether the regulation of RPS7 ubiquitination by OTUD6B is conserved in mammalian cells, including human cells.
3. In Fig. 2, the IP-MS workflow used to identify the OTUD6 interactome was not clearly presented. Additionally, it is unclear what the negative controls were in this experiment. While using an inactive form of OTUD6 for IP-MS is a good idea, the

selection of this inactive form should be better justified.

4. Figure 4 presents an MS analysis of the ubiquitinated proteome, which was useful in identifying RPS7. However, the other identified proteins from this analysis should also be presented in the supplemental figures/tables. Additionally, including a pathway/network analysis of these proteins would provide a more comprehensive understanding of the potential pathways and networks involved in.

5. In Fig. 7, the study shows that under MMS stress conditions, there are changes in the levels of OTUD6 and RPS7 monoubiquitination. However, it is unclear whether the levels of upstream proteins, such as RACK1 and E3 ligases CNOT4 and RNF10, are also altered under these conditions. Further investigation (e.g., a proteomic analysis) into the levels and potential regulation of these upstream proteins could provide a more complete understanding of the mechanisms by which RPS7 ubiquitination and deubiquitination are regulated under different stress conditions.

Author Rebuttal letter:

Reviewer 1.

Major Comments:

1) For Figure 6B, the IP of RPL11-GFP does not reveal detectable ubiquitinated RPS7 in the IP, so it is not a good control to suggest that OTUD6 does not associate with 80S because OTUD6 is absent in the IP (see comment 2 where authors now show that ubiquitinated RPS7 is present in 80S)

We now include a longer exposure that shows ubiquitinated RPS7 in the RPL11.GFP but not the control immunoprecipitates (6B is now 7C).

2) Why do the authors argue that the increase of ubiquitination of RPS7 in the 80S in the OTUD6 C183R mutant suggests that the ubiquitination of RPS7 prevents RPS7-Ubi incorporation into the translating ribosome (Figure 6C)? Does this result per se actually argue for this?

We agree that there exist additional possible mechanisms. We now make the observation without providing a specific mechanistic interpretation. (Fig 6C is now 7D)

3) If I understand correctly, the authors propose that the presence of one transgene can alter the level of RPS7 ubiquitination dependent upon the transgene content of non-optimal codons. Can the authors justify this interpretation? Is the transgene so much overexpressed that it can have an effect visible above all the ongoing translation in the cell. Also, the effect shown on Figure 6D is not convincing.

This is what the data shows: the presence of one transgene with 50% nonoptimal codons increases the monoubiquitination of RPS7 (now Fig 6C). Both the optimal (GFP0D) and nonoptimal (GFP54C3) transgenes are driven off of the Ubi ubiquitin promoter that expresses at high levels in all cells, so a detectable signal may not be so surprising. Indeed, the original publication for the optimal and non-optimal reporters showed that GFP expression off the reporters was detectable in live whole organisms and in dissected tissue (<https://doi.org/10.7554/eLife.76893>). Moreover the effect on RPS7 ubiquitination is supported statistically: in the figure we show one representative western blot that is one biological replicate. Each dot in the accompanying graphs are separate biological replicates: we measured a clear increase in RPS7.Ub in GFP54C3', compared to GFP0D, across biological replicates.

4) What is the evidence mentioned in the discussion that there are other routes to RPS7 deubiquitination according to the known routes for mono-ubiquitination? This is not clear.

We removed this sentence from the discussion.

5) If there is less re-initiation of 40S, and less 40S associated with the mRNA, should not also then less 60S rejoin the mRNA and then be less 60S?

Yes, but we are using 60S controls to look at 40S ribosomes. An 80S ribosome is a 1:1 ratio of 40S to 60S: any other 40S ribosome will be from 43S/48S and terminating 40S ribosomes (was Fig 6E, now Fig 7E; mRNA capture and western detection of ribosomal components.)

6) The authors use many times RPS7-Ubi relative to overall Ubi for quantification, and this is really a very strange quantification. It should be as in Figure 7E, relative to RPS7 levels. Whether changes are then similar or not to overall ubiquitination level changes should be a separate measurement. I am also curious about how overall ubiquitination is evaluated. I could not find the description in the methods. As it stands, I am not convinced by these ratios (Figure 4D, 5A and 5B)

We redid the measurements to be relative to total RPS7 levels. The conclusions are unchanged for 4D and 5A (these panels are now 5A and 6A, respectively) For 5B (now 6B), MMS reduces RPS7.Ub as

measured previously, however there is now no effect of MMS on RPS7.Ub in OTUD6.C183A mutants. This updated finding suggests that the effects of MMS on RPS7 deubiquitination is via OTUD6. We also showed that total ubiquitination levels are not changed by MMS treatment (supplemental S5F).

Minor comments:

1) Do the authors suggest that their higher affinity OTUD6 C183R is dominant negative since it is MMS sensitive in the heterozygote?

C183A is likely higher affinity; yes, it is behaving as a dominant negative. The predicted lower affinity C183R behaves as a recessive loss of function.

2) The authors mention that no poly-Ubi RPS7 is detectable in the OTUD6 mutants but they do not show entire blots, this should be made available in supplementary data.

Added Figure 4D, the full blot for Figure 4C.

3) The staining in Figure 1G expressed to be in nucleus and nucleolus should be indicated (maybe arrow for an example of this), because this is not visible on the figure.

We included higher resolution and increased gain micrograph in Supplementary Figure 1D.

4) On Figure 4E, panel E, what are L and S (it seems high and low exposure, but there is no legend for this). Same comment for Figure S3D.

We now include L: Long exposure and S: Short exposure in all of the relevant figure legends. (Fig 4E is now 5B)

5) Figure 7A (and others): are the dots multiple experiments (biological repeats) or multiple repeats of the same samples (a question that is generally relevant for the quantifications).

Biological replicates - true for all bar graph data in the manuscript. This is stated in the Statistical Analysis section of the Materials and Methods, and now also in the legend for Figure 1. (Fig 7A is now 8A)

6) The ubiquitination of CNOT4 has never actually been shown to occur during 80S assembly during initiation or when non-optimal codons are encountered, nor has it been shown that this is coupled to mRNA deadenylation. These are all interpretations of observed data and should be presented as such (it has been proposedâ[.]).

We adopted the phrasing suggested by the reviewer. The relevant sentences are:

Results: "CNOT4 is proposed to ubiquitinate RPS7 during 80S assembly at the start codon, and when nonoptimal codons are encountered, a process that can be coupled to mRNA deadenylation by the CCR4-NOT complex 30,31."

Results: "RPS7 is proposed to be monoubiquitinated by CNOT4 in the presence of nonoptimal codons in mRNAs 7,31."

Discussion: "CNOT4/NOT4 is proposed to ubiquitinate RPS7 on the 48S and on the 80S ribosome 7,28,30,31"

."

Reviewer #2 (Remarks to the Author):

The authors did not examine the effect of MMS treatment on protein translation at all in this manuscript. We did this experiment - MMS dramatically reduces protein translation in Drosophila (Supplementary Figure 5G), matching what is observed in a variety of other organisms.

Other minor points are also listed below.

1. There is no data to speculate how the mutations in DIS3 suppress MMS sensitivity in the OTUD6 mutant. Is it possible that the mutations in DIS3 reduce RPS7 monoubiquitylation?

We decided to keep our focus on the OTUD6-RPS7-protein translation interaction and did not pursue this intriguing possibility.

2. It would be better to describe clearly which E2s and E3s are able to suppress the phenotype in the OTUD6 mutant in Figure 2D and supplementary figure 2.

Now rewritten: "Three E3 ligases, CNOT4, RNF10, and RNF123/KPC1 suppressed OTUD6C183A sensitivity to MMS treatment (Figure 2E-G, Supplementary Figure 2A-G). A fourth, FBXW7/CDC4, showed weaker but significant suppression of OTUD6C183A MMS sensitivity (Supplementary Figure 2A). All others either did not suppress sensitivity or were sensitive on their own."

3. At lines 149 and 150, the authors claimed that they did not detect bands that would be consistent with polyubiquitination of RPS7. However, this reviewer cannot determine if the authors' claim is correct with the current Fig. 4C because upper areas of the membranes were cut off.

Added the full blot as Figure 4D, the full blot for Figure 4C.

4. Have the authors examined whether overexpression of RNF10 and RACK1 increase in monoubiquitylation of RPS7 and decrease in protein synthesis rates?

We have not, because we would need to make and then validate the tools (none exist in *Drosophila*, as opposed to mammalian experimental setups). This is an intriguing question that can help identify the roles of putative upstream E2/E3s.

5. It is overstating to describe that flies may tune down protein translation by decreasing OTUD6 as they age at lines 229 and 230. The authors should examine whether overexpression of OTUD6 recovers protein translation even in aged flies.

We show in Figure 5A that OTUD6 overexpression increases global protein translation.

6. In Figure 6B, this reviewer does not understand why monoubiquitinated RPS7 was not detected because RPS7 in this figure was purified from 80S ribosomes.

Now Figure 7C: we added a longer exposure that shows monoubiquitinated RPS7 is detected.

7. At lines 224 and 225, the authors supposed that increase in OTUD6 abundance led to decrease in monoubiquitylated RPS7 under MMS treatment. However, decrease in monoubiquitylated RPS7 occurred in the OTUD6 C183A mutant, implying that decrease in monoubiquitylated RPS7 is independent of OTUD6. The authors also should examine whether MMS treatment affects global protein translation rate or not in Figure 7A.

At the request of Reviewer 1 (point 6), we now compare to total RPS7 levels instead of to total ubiquitin levels. With this change we observe a decrease in RPS7 monoubiquitylation in controls treated with MMS but not in the OTUD6 C183A mutant. Thus our current data is consistent with MMS treatment causing an increase in OTUD6, leading to a decrease in RPS7 monoubiquitylation.

We also did the requested experiment: MMS dramatically reduces global protein translation rate in *Drosophila* (now presented as Supplementary Figure 5G), as has been observed in a variety of other organisms. Based on current understanding, we suspect that MMS acts through another pathway to decrease global protein translation, perhaps via the integrated stress response (ISR) pathway as has been recently reported. Upregulation of OTUD6 and increased RPS7 deubiquitylation might be a compensatory pathway to counteract the ISR effect on translation. This is an interesting area of future research. (Fig 7A is now 8A)

Reviewer #3 (Remarks to the Author):

1. To definitively demonstrate the role of RPS7 ubiquitination, it would be important to identify the precise site(s) of ubiquitination and perform mutational analysis by substituting the Lysine residue(s) with Arginine. Then it would be possible to monitor the translational activity upon the mutation or analyze the growth under stress conditions. Some known human RPS27 ubiquitination sites may be found in the Phosphosite database (<https://www.phosphosite.org/proteinAction.action?id=6553&showAllSites=true>)

Thank you for requesting this informative experiment. We created RPS7 that is mutant for two adjacent lysine residues, including the conserved site that is ubiquitinated in yeast eS7A, by CRISPR of the endogenous gene (RPS7(K2R)). Homozygotes are fertile and viable and do not exhibit any Minute phenotypes, so ribosome function is intact. RPS7 monoubiquitination in RPS7(K2R) and OTUD6(C183A),RPS7(K2R) is undetectable (Figure 4E,F). MMS sensitivity was increased and global protein translation was reduced in RPS7(K2R) (Figure 6D,E).

2. Is the ubiquitination regulation of RPS7 by OTUD6 conserved in mammalian cells? Considering that OTUD6B mutations are associated with human intellectual disability, it is highly relevant to show or at least discuss whether the regulation of RPS7 ubiquitination by OTUD6B is conserved in mammalian cells, including human cells.

We added the following to the Discussion: "RPS7 monoubiquitination was increased in catalytically inactive OTUD6 flies, and it appeared to be the only detectable form of RPS7 that immunoprecipitated with tagged catalytically inactive OTUD6. Consistent with our findings, the yeast ortholog of OTUD6, OTU2, deubiquitinates yeast RPS7/eS7A on the 40S ribosomal subunit 49â51. The lysine on yeast RPS7/eS7A that is deubiquitinated by OTU2 is conserved in flies and it is deubiquitinated by OTUD6, and the lysine for ubiquitination is also conserved in human RPS7. While ubiquitination/deubiquitination of human RPS7 at the conserved lysine is not yet reported in mammals, OTUD6B was recovered on the human 40S ribosomal subunit in kinase-dead RIOK1 cells 52."

We decided to keep our experimental focus on the mechanism in *Drosophila*. A thorough study of the activities of OTUD6B and its splice isoforms is essential to understand their roles on mammals.

3. In Fig. 2, the IP-MS workflow used to identify the OTUD6 interactome was not clearly presented. Additionally, it is unclear what the negative controls were in this experiment. While using an inactive form of OTUD6 for IP-MS is a good idea, the selection of this inactive form should be better justified.

We used anti-FLAG magnetic beads to immunoprecipitate proteins from OTUD6.C183A.FLAG.HA vs OTUD6.C183A untagged fly heads, as described in the Materials and Methods section "Antibody Coupling to Beads and Co-immunoprecipitation." This design ensured the most similar cellular environment and allowed us to select away from non-specific binding to the immunoprecipitation beads. We added the label "FLAG IP: OTUD6C183A.FLAG.HA vs. OTUD6C183A" to Figure 2A, and we added the following sentence to the Results (1st sentence included here for context): "To identify potential OTUD6 substrates, we performed co-immunoprecipitation (co-IP) followed by mass spectrometry (MS) with OTUD6C183A.FLAG.HA as compared to untagged OTUD6C183A from *Drosophila* heads. This design allowed us to segregate OTUD6C183A interactors from non-specific binding to the immunoprecipitation matrix."

4. Figure 4 presents an MS analysis of the ubiquitinated proteome, which was useful in identifying RPS7. However, the other identified proteins from this analysis should also be presented in the supplemental figures/tables. Additionally, including a pathway/network analysis of these proteins would provide a more comprehensive understanding of the potential pathways and networks involved in.

The data is in Supplementary Data files 1,2, and 3. We decided to leave network analysis to any curious readers, since we want to keep our focus on the OTUD6-RPS7-protein translation relationship for this manuscript.

5. In Fig. 7, the study shows that under MMS stress conditions, there are changes in the levels of OTUD6 and RPS7 monoubiquitination. However, it is unclear whether the levels of upstream proteins, such as RACK1 and E3 ligases CNOT4 and RNF10, are also altered under these conditions. Further investigation (e.g., a proteomic analysis) into the levels and potential regulation of these upstream proteins could provide a more complete understanding of the mechanisms by which RPS7 ubiquitination and deubiquitination are regulated under different stress conditions.

We agree this is interesting, because it could indicate additional regulatory steps for tuning global protein translation through the OTUD6-RPS7 mechanism. We found no difference in RACK1 abundance with MMS treatment (1.00 control vs 1.072 +MMS, normalized to alpha-tubulin, n=3 biological replicates, P=0.5871 two-tailed t-test). We opted to not include this data, because we have no reagents in *Drosophila* to similarly quantify CNOT4 and RNF10, to form a complete answer. (Fig 7 is now Fig 8)

Version 1:

Reviewer comments:

Reviewer #1

(Remarks to the Author)

The revised manuscript is very much improved and has addressed most of my concerns appropriately.

I have only minor points remaining:

1) it remains difficult to see the OTUD6 staining in the nucleus and nucleolus

2) page 9, line 117, and page 7, line 193, this description about when CNOT4 ubiquitinates RPS7 and the link to deadenylation is better than before because it is now not an assertion, but what has really been shown is what ribosomes have bound CNOT4. Maybe this could be tuned down further.

3) page 8, line 207, "further" is mis-spelt

4) page 8, line 210, I don't think that it can be said that RPS7 is monoubiquitinated by a variety of E3 ligases, this has only been shown for CNOT4. The others may play a regulatory role (for instance on CNOT4 itself, but other possibilities exist)

5) Do the authors think that it is an issue to do experiments with cycloheximide that can compete with CNOT3 binding to the ribosome E site (and thus maybe CNOT4) and purification of mRNAs via oligo-dT beads if there is a possible link between RPS7Ubi-deUbi and deadenylation? Should something be mentioned?

Reviewer #2

(Remarks to the Author)

The revised manuscript by Villa and colleagues is improved over the original manuscript. It is obvious that the authors made an effort to address the criticisms that were made during the first review. However, there are some points that this reviewer would like to confirm about their revised manuscript and their responses as described below.

Response to point #5: If the authors want to claim their speculation about the relationship between OTUD6 protein abundance, protein translation and lifespan, it may be necessary to investigate whether lifespan is shortened in OTUD6-overexpressing mutants.

There is no data showing that "we were unable to detect association of OTUD6 with RNA" in line 246.

It should be noted which data was used as a reference for the protein levels of nonubiquitinated RPS7 in the figure legend for supplementary figure 4.

It is not clear which band represents RPL11 in supplementary figure 6A. Distribution patterns of RPL11 are quite different between Figure 7B and supplementary figure 6A.

Reviewer #3

(Remarks to the Author)

My concerns have been addressed. In particular, the site-directed mutagenesis experiment of ubiquitination sites is convincing. The manuscript is ready for publication.

Author Rebuttal letter:

Reviewer #1 (Remarks to the Author):

The revised manuscript is very much improved and has addressed most of my concerns appropriately. Thank you.

I have only minor points remaining:

1) it remains difficult to see the OTUD6 staining in the nucleus and nucleolus.

Response: We further enhanced the gain. The cytoplasmic expression is so strong that overwhelms the nucleolar signal.

2) page 9, line 117, and page 7, line 193, this description about when CNOT4 ubiquitinates RPS7 and the link to deadenylation is better than before because it is now not an assertion, but what has really been shown is what ribosomes have bound CNOT4.

Maybe this could be tuned down further.

These are the two sentences:

"CNOT4 is proposed to ubiquitinate RPS7 during 80S assembly at the start codon, and when nonoptimal codons are encountered, a

process that can be coupled to mRNA deadenylation by the CCR4-NOT complex 30,31."

"RPS7 is proposed to be monoubiquitinated by CNOT4 in the presence of nonoptimal codons in mRNAs 7,31."

We changed the first to:

"CNOT4 is proposed to ubiquitinate RPS7 during 80S assembly at the start codon, and when nonoptimal codons are encountered, a

process that can be coupled to mRNA stability by the CCR4-NOT complex 30,31."

We think that these two sentences will help direct interested researchers to the cited papers.

3) page 8, line 207, "further" is mis-spelt

Thank you

4) page 8, line 210, I don't think that it can be said that RPS7 is monoubiquitinated by a variety of E3 ligases, this has only been

shown for CNOT4. The others may play a regulatory role (for instance on CNOT4 itself, but other possibilities exist) In Fig 6A, we show that mutation of RNF10, RACK1, and CNOT4 decrease RPS7 monoubiquitination, and in Fig 2C, 2E, and 2F we show that OTUD6 likely acts downstream of RNF10, RACK1, and CNOT4. So RNF10 and RACK1 behave like CNOT4 in these assays. However, we agree that direct biochemical evidence for RNF10 and RACK1 ubiquitination of RPS7 does not exist. The sentence in question is now changed from

"Together, these results suggest that RPS7 monoubiquitination by a variety of E3 ligases and subsequent deubiquitination by OTUD6 is critical for maintaining resilience to alkylation stress and for the promotion of protein translation."

to

"Together, these results suggest that RPS7 monoubiquitination by a variety of E3 ligases, directly or indirectly, and subsequent deubiquitination by OTUD6 is critical for maintaining resilience to alkylation stress and for the promotion of protein translation. Direct ubiquitination of RPS7 by RNF10 and RACK1 is not yet demonstrated."

5) Do the authors think that it is an issue to do experiments with cycloheximide that can compete with CNOT3 binding to the ribosome E site (and thus maybe CNOT4) and purification of mRNAs via oligo-dT beads if there is a possible link between RPS7Ubi-deUbi and deadenylation ? Should something be mentioned?
We did not add cycloheximide to mRNA capture experiment.

Reviewer #2 (Remarks to the Author):

The revised manuscript by Villa and colleagues is improved over the original manuscript. It is obvious that the authors made an effort to address the criticisms that were made during the first review.
Thank you.

However, there are some points that this reviewer would like to confirm about their revised manuscript and their responses as described below.

Response to point #5: If the authors want to claim their speculation about the relationship between OTUD6 protein abundance, protein translation and lifespan, it may be necessary to investigate whether lifespan is shortened in OTUD6-overexpressing mutants.

We are aware that causation cannot be inferred from the experiments we provide. We included the words "might" and "potential" in our revised manuscript to indicate this, as detailed below. We also agree that further experiments are needed to establish or refute a causal role. A brief reminder: OTUD6 mutant flies live longer, OTUD6 protein levels are decreased with age, and decreased OTUD6 is associated with reduced global protein translation.
The relevant sentences are

Results: "OTUD6 protein abundance was lower in old flies, suggesting that flies might tune down protein translation by decreasing OTUD6 levels as they age."

Discussion: "OTUD6 protein levels declined with age, providing a potential mechanism for age-dependent decrease in protein translation levels."

We changed this last sentence to: "OTUD6 protein levels declined with age, providing a potential mechanism for age-dependent decrease in protein translation levels; additional experiments are needed."

There is no data showing that we were unable to detect association of OTUD6 with RNA in line 246. While we have the blots showing this, we decided instead to remove the sentence.

It should be noted which data was used as a reference for the protein levels of nonubiquitinated RPS7 in the figure legend for supplementary figure 4.

Thank you for catching this. We now indicate that it is the measurements taken in Fig 5B.

It is not clear which band represents RPL11 in supplementary figure 6A. Distribution patterns of RPL11 are quite different between

Figure 7B and supplementary figure 6A.

We now provide molecular weights beside the western panels, to help indicate the appropriate RPL11 bands. The gel used for the supplementary figure was at a higher percentage, and so the non-specific bands are presented differently.

Reviewer #3 (Remarks to the Author):

My concerns have been addressed. In particular, the site-directed mutagenesis experiment of ubiquitination sites is convincing. The manuscript is ready for publication.

Thank you.
